# Scaling Up Active Testing to Large Language Models

**Gabrielle Berrada**[1]*  **Jannik Kossen**[1]*  **Freddie Bickford Smith**[2]  **Muhammed Razzak**[1]

**Yarin Gal**[1]  **Tom Rainforth**[2]†

[1] OATML, Department of Computer Science, University of Oxford
[2] Department of Statistics, University of Oxford

## Abstract

Active testing enables label-efficient evaluation of predictive models through careful data acquisition, but it can pose a significant computational cost. We identify cost-saving measures that enable active testing to be scaled up to large language models (LLMs). In particular we show that the surrogate model used to guide data acquisition can be constructed cheaply using in-context learning, does not require updating within an active-testing loop, and can be smaller than the target model. We even find we can make good data-acquisition decisions without making predictions with the target model. As a result we are able to achieve much more accurate evaluations of LLM performance relative to using randomly acquired data. We additionally introduce a bootstrap estimator of evaluation error, which we show to be a useful indicator of how well active testing is working within a single run.

## 1  Introduction

Evaluating frontier models is becoming more expensive as they become more sophisticated (Burden, 2024; OpenAI, 2023). At the same time, with new models arriving in quick succession, and ever-present scope for data to leak from evaluations to training (Ganguli et al, 2023), the evaluation problem is a dynamic one: it requires ongoing, adaptive gathering of new evaluation data.

Active testing (Kossen et al, 2021, 2022) is an attractive solution to this problem. Motivated by the observation that some labels are more informative than others about the behaviour of a target model, it involves carefully deciding which test inputs to acquire labels for. The foundation for data acquisition is a surrogate model of the test-time label distribution, which is typically updated on the labels acquired during testing. Making a data-acquisition decision requires making predictions with the surrogate model and often also the target model. The computational cost of this process is often justifiable in light of high labelling costs, but it limits the scope of active testing's applicability.

In this work we show how active testing can be scaled up to large language models (LLMs), allowing us to more effectively evaluate them with limited labelling budgets. We begin by highlighting three computational bottlenecks in the predominant approach to active testing: updating the surrogate model as labels are acquired, making predictions with the surrogate model, and making predictions with the target model. The first of these is particularly important: it has traditionally involved repeatedly running gradient-based training on the acquired test data within an active-testing loop.

We then identify simple and surprisingly effective ways to address each of these key bottlenecks. First, we strip back the training of the surrogate model to a single step of in-context learning (Brown et al, 2020) on a small amount of initial test data, removing the need for repeated gradient-based training. Second, we show it is possible to use a surrogate model that is smaller than the target model,

---

*Equal contribution

†Correspondence to rainforth@stats.ox.ac.uk.

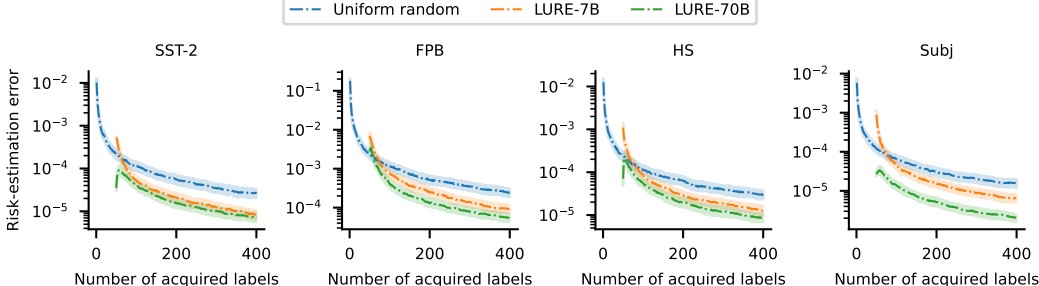

**Figure 1** Our proposed active-testing approach enables low-error estimates of the risk (expected predictive loss) of a large language model (here the 70B version of Llama 2) on four text-classification problems (SST-2, FPB, HS and Subj) while only using a small label budget. We compare uniform-random testing against active testing (LURE), using either a 7B or 70B surrogate model with in-context learning to guide data acquisition.

reducing the cost of surrogate-model predictions. Third, we show we can even forgo making any predictions with the target model, relying solely on the surrogate model for data acquisition. Together these changes make active testing dramatically cheaper, enabling it to be scaled up to LLMs.

Empirically we find our approach can substantially improve over the common practice of acquiring data uniformly at random (Figure 1). More specifically it can estimate the risk (expected predictive loss over test examples) of LLMs with an estimation error typically 25% to 50%—and sometimes up to 80%—lower than that obtained through uniform-random data acquisition. Our work therefore represents significant progress in accurately and dynamically evaluating LLMs, as well as in more generally increasing the scope for active testing to be used in practical settings.

To further improve the real-world applicability of active testing, we address the challenge of judging how well it is working on a given problem. In research we can assess active testing by running it multiple times and comparing its risk estimates to a known true risk. But practical deployment involves running active testing only once, without knowledge of the true risk, making it hard to gauge active-testing performance. To help address this we derive a bootstrap estimator (Efron, 1979) of the risk-estimation error. In experiments we find our approximate 95% confidence intervals contain the true risk-estimation error 88% of the time, suggesting our estimator can be a useful diagnostic tool.

## 2 Background

Our aim is to evaluate a fixed target model, $f$, mapping inputs $x \in \mathcal{X}$ to outputs $y \in \mathcal{Y}$. We formalise evaluation as estimating a form of frequentist risk (Berger, 1985), namely an expected predictive loss of the form $R = \mathbb{E}_{p_{\text{eval}}(x,y)}[\ell(f(x), y)]$, where $p_{\text{eval}}$ represents a reference system used as a source of ground truth, and $\ell$ denotes a loss function that represents the consequences of predictive errors. As in past work (Kossen et al, 2021, 2022), we consider a pool-based setting (Lewis & Gale, 1994) where we have access to a pool of $N$ unlabelled test inputs, $\mathcal{D}_{\text{pool}} = \{x_i\}_{i=1}^{N}$, but acquiring a label, $y \sim p_{\text{eval}}(y|x)$, for any given $x$ is costly, so we can only afford to acquire $M < N$ labels.

### 2.1 Uniform-random sampling

A simple estimator of $R$ uses uniform-random samples from the pool:

$$\hat{R}_{\text{unif}} = \frac{1}{M} \sum_{m=1}^{M} \ell(f(x_{i_m}), y_{i_m}),$$

where $i_{1:M} \sim \text{Uniform}(\{1, 2, \ldots, N\}, M)$ are indices sampled without replacement. If the pool was constructed by sampling $x_i \sim p_{\text{eval}}(x)$ then $\hat{R}_{\text{unif}}$ is known as the subsample empirical risk and is an unbiased estimator of $R$. However, it will typically have high variance for small values of $M$.

### 2.2 Sampling-based active testing

Active testing deals with how to more carefully select the $M$ inputs for labelling to produce a more accurate estimate of $R$. This can be achieved through either a sampling-based approach or an

| **Algorithm 1** Sampling-based active testing | **Algorithm 2** Interpolation-based active testing |
|---|---|
| **input** Target model, $f$; loss function, $\ell$; acquisition function, $a$; training set, $\mathcal{D}_{\text{train}}$; pool set, $\mathcal{D}_{\text{pool}}$; label budget, $M$ | **input** Target model, $f$; loss function, $\ell$; acquisition function, $a$; training set, $\mathcal{D}_{\text{train}}$; pool set, $\mathcal{D}_{\text{pool}}$; label budget, $M$ |
| 1: Compute $f(x_j)$ for all $x_j \in \mathcal{D}_{\text{pool}}$ | 1: Compute $f(x_j)$ for all $x_j \in \mathcal{D}_{\text{pool}}$ |
| 2: Set $\mathcal{D}_{\text{test}} = \emptyset$ | 2: Set $\mathcal{D}_{\text{test}} = \emptyset$. |
| 3: **for** $m \in (1, 2, \dots, M)$ **do** | 3: **for** $m \in (1, 2, \dots, M)$ **do** |
| 4:   Train $\pi_m$ (e.g., on $\mathcal{D}_{\text{train}} \cup \mathcal{D}_{\text{test}}$) | 4:   Train $\pi_m$ (e.g., on $\mathcal{D}_{\text{train}} \cup \mathcal{D}_{\text{test}}$) |
| 5:   Compute $a_m(x_j)$ for all $x_j \in \mathcal{D}_{\text{pool}}$ | 5:   Compute $a_m(x_j)$ for all $x_j \in \mathcal{D}_{\text{pool}}$ |
| 6:   Sample $i_m \sim q_m(i)$ | 6:   Select $i_m = \arg\max_j a_m(x_j)$ |
| 7:   Sample $y_{i_m} \sim p_{\text{eval}}(y|x_{i_m})$ | 7:   Sample $y_{i_m} \sim p_{\text{eval}}(y|x_{i_m})$ |
| 8:   Set $\mathcal{D}_{\text{test}} \leftarrow \mathcal{D}_{\text{test}} \cup \{(x_{i_m}, y_{i_m})\}$ | 8:   Set $\mathcal{D}_{\text{test}} \leftarrow \mathcal{D}_{\text{test}} \cup \{(x_{i_m}, y_{i_m})\}$ |
| 9: **end for** | 9: **end for** |
| 10: Compute $\hat{R}_{\text{LURE}}$ | 10: Compute $\hat{R}_{\text{ASE}}$ |
| **output** Risk estimate, $\hat{R}_{\text{LURE}}$ | **output** Risk estimate, $\hat{R}_{\text{ASE}}$ |

interpolation-based approach. In sampling-based active testing we use Monte Carlo estimators of $R$, similar to $R_{\text{unif}}$ but with pool indices sampled from a non-uniform distribution. At acquisition step $m$ we sample pool input $x_i$ with probability proportional to $a_m(x_i) \in \mathbb{R}^+$ where $a_m$ is an acquisition function that measures a notion of how useful we expect the label for a given input to be (the $m$ subscript denotes that it can depend on all data available at step $m$, including the target model's training data). That is, we sample $i_m \sim q_m(i)$ where $q_m(i) = a_m(x_i) / \sum_{x_j \in \mathcal{D}_{\text{pool}}} a_m(x_j)$.

Farquhar et al (2021) showed that naive Monte Carlo with $i_m \sim q_m(i)$ is biased and, to address this, introduced the levelled unbiased risk estimator (LURE):

$$\hat{R}_{\text{LURE}} = \frac{1}{M} \sum_{m=1}^{M} v_m \ell(f(x_{i_m}), y_{i_m}), \quad v_m = 1 + \frac{N-M}{N-m}\left(\frac{1}{(N-m+1)q_m(i_m)} - 1\right). \quad (1)$$

Given both $\hat{R}_{\text{unif}}$ and $\hat{R}_{\text{LURE}}$ are unbiased, any advantage that $\hat{R}_{\text{LURE}}$ brings comes from variance reduction through a well-designed acquisition function, $a_m$. The optimal acquisition function is $a_m^*(x) = \mathbb{E}_{p_{\text{eval}}(y|x)}[\ell(f(x), y)]$, the expected loss under $p_{\text{eval}}(y|x)$. Since this is unknown, Kossen et al (2021) proposed approximating $p_{\text{eval}}(y|x)$ with a surrogate model, $\pi_m(y|x)$, which is typically trained on the acquired test labels along with the target model's training data. This surrogate model then allows us to define a practical acquisition function:

$$a_m(x) = \mathbb{E}_{\pi_m(y|x)}[\ell(f(x), y)]. \quad (2)$$

If $f(x) = p_f(y|x)$ and $\ell(\hat{p}, y) = -\log \hat{p}(y)$ then the acquisition function becomes $a_m(x) = \text{H}[\pi_m(y|x) \| p_f(y|x)]$, the cross entropy between the surrogate and target models. Intuitively we can understand this acquisition function as measuring the disagreement between the surrogate and target models, leading us to acquire labels for the inputs where the two models' predictions differ the most.

### 2.3   Interpolation-based active testing

Kossen et al (2022) introduced an alternative approach to active testing that, like the sampling-based approach, uses a surrogate model to guide data acquisition, but also uses it to estimate the risk. Their active surrogate estimator (ASE) is

$$\hat{R}_{\text{ASE}} = \frac{1}{N} \sum_{x_i \in \mathcal{D}_{\text{pool}}} \mathbb{E}_{\pi_m(y|x_i)}[\ell(f(x_i), y)]. \quad (3)$$

Here the risk is not directly estimated using labels acquired from $y \sim p_{\text{eval}}(y|x)$ but instead using labels simulated from the surrogate model. The goal of data acquisition is then to improve the surrogate model in a way that leads to a more accurate estimate of the risk. Kossen et al (2022) approached this using an acquisition function called the expected weighted disagreement (XWED).

# 3 Scaling up active testing

With an understanding of standard approaches to active testing, we can now identify and address barriers to scaling up to LLMs. We focus on three key computational bottlenecks: training the surrogate model, making predictions with the surrogate model, and making predictions with the target model. Our aim is to reduce the cost of these steps while maintaining the efficacy of active testing.

## 3.1 Surrogate-model training

We argue that the top priority for reducing the cost of data acquisition is to rethink the training of the surrogate model. Existing approaches involve repeatedly running gradient-based training on the data acquired during testing, typically combined with the target model's training data. This can be very expensive, especially when working with large datasets and surrogates, as we are focusing on here.

To avoid this expense we suggest a stripped-back approach: construct the surrogate model using in-context learning (Brown et al, 2020) on a small amount of randomly acquired test data, then keep it fixed. This reduces the cost of surrogate-model training to essentially the minimum possible.

This design decision has important implications for the relative merits of sampling-based and interpolation-based active testing. We are using a relatively crude surrogate model to reduce computational cost. This affects the sampling-based estimator solely through the data-acquisition step (and related corrective weights, $v_m$), whereas it affects the interpolation-based estimator not only through data acquisition but also much more directly through the expectation over $\pi_m(y|x)$ in Equation 3. Due to this difference in sensitivity between the approaches, we default to the sampling-based estimator for this choice of surrogate model, and demonstrate in Section 5 that this is well-justified empirically.

## 3.2 Surrogate-model predictions

Another cost is that of making predictions with the surrogate model. In sampling-based active testing these predictions arise when computing the acquisition distribution, $q_m(i)$, and in interpolation-based active testing they additionally arise when computing the risk estimate.

Our choice to use a fixed surrogate model, $\pi_0$, automatically leads to a significant reduction in how many of these predictions are required: we only need to compute $\pi_0(y|x_j)$ once for each $x_j \in \mathcal{D}_{\text{pool}}$. On top of this we can make each individual forward pass cheaper by using a smaller surrogate model.

## 3.3 Target-model predictions

The third cost we aim to reduce is that of making predictions with the target model. The standard versions of both sampling-based and interpolation-based active testing require computing $f(x_j)$ for each $x_j \in \mathcal{D}_{\text{pool}}$, which is especially expensive for large target models and pool sets.

This is unavoidable in the interpolation-based approach due to the construction of $\hat{R}_{\text{ASE}}$. But in the sampling-based approach there is scope for an approximation that can reduce the number of target-model predictions from $N$ to $M$, where often $M \ll N$. In particular, when computing the acquisition distribution, $q_m(i)$, we can use the surrogate model to approximate the target model's predictions on the pool set. This leads to a new acquisition function,

$$\hat{a}_m(x) = \mathbb{E}_{\pi_m(y|x)}[\ell(\psi[\pi_m(\cdot|x)], y)],$$

where $\psi$ denotes an operation that accounts for the fact that the first argument of $\ell$ might not be a probability distribution (e.g., if $\ell(\hat{y}, y) = \|\hat{y} - y\|_2^2$, then we require something like $\psi[\pi_m(\cdot|x)] = \mathbb{E}_{\hat{y} \sim \pi_m(\cdot|x)}[\hat{y}]$). This acquisition function can be seen as an approximation to $a_m(x)$ in Equation 2. If $\ell(\hat{p}, y) = -\log \hat{p}(y)$ then $\hat{a}_m(x) = \mathrm{H}[\pi_m(y|x)]$, the predictive entropy of the surrogate model.

## 3.4 Active testing for dataset curation

The use of the surrogate model to approximate the target model during data acquisition also makes it possible to use active testing for dataset curation, where all the inputs in the pool set are already labelled and the goal is to reduce the number of target-model predictions required for evaluation (Maynez et al, 2023; Polo et al, 2024; Saranathan et al, 2024; Vivek et al, 2024). This can be viewed

as a variation on standard active testing where here we want to use the surrogate model to select $M < N$ input-label pairs with which to evaluate the target model. The acquisition function in this case can depend on the labels as well as the inputs:

$$a_m(x, y) = \ell(\psi[\pi_m(y|x)], y). \tag{4}$$

If we use the logarithmic loss as before, this recovers the negative log likelihood of the surrogate.

## 4 Estimating risk-estimation error

Aside from its scalability, active testing poses a practical challenge in assessing the quality of the risk estimates it produces. In research we can construct repeatable simulations of active testing and record ground-truth risk-estimation errors across repeat runs. But practical deployment of active testing involves only a single run, without knowledge of the true risk. This leaves practitioners unaware of how well their implementation is working, making it harder to justify using active testing.

We propose addressing this using a novel estimator of the mean squared error (over possible sequences of acquired indices, $i_{1:K} \sim q(i_{1:K})$, where $K \le M$) of $\hat{R}_{\text{LURE}}$. For a generic risk estimator, $\hat{R}(i_{1:K})$, we can decompose the mean squared error into squared bias and variance:

$$\text{MSE}(\hat{R}) = \mathbb{E}_{q(i_{1:K})}[(\hat{R}(i_{1:K}) - R)^2] = \underbrace{(\mathbb{E}_{q(i_{1:K})}[\hat{R}(i_{1:K})] - R)^2}_{\text{Bias}(\hat{R})^2} + \underbrace{\mathbb{V}_{q(i_{1:K})}[\hat{R}(i_{1:K})]}_{\text{Var}(\hat{R})}. \tag{5}$$

Since $\text{Bias}(\hat{R}_{\text{LURE}}) = 0$, our task is to estimate $\text{Var}(\hat{R}_{\text{LURE}}) = \text{MSE}(\hat{R}_{\text{LURE}})$ from a single sequence of acquired indices, $i_{1:K}$. Letting $L = (v_m \ell(f(x_{i_m}), y_{i_m}))_{m=1}^{K}$ denote the first $K$ reweighted losses used in Equation 1, a bootstrap estimator (Efron, 1979) of the risk, $R$, is

$$\hat{R}_{\text{boot}} = \frac{1}{K} \sum_{m=1}^{K} L_{j_m}, \quad j_m \sim \text{Uniform}(1, 2, \ldots, K).$$

If we have $B$ bootstrap estimates, $(\hat{R}_{\text{boot}}^b)_{b=1}^{B}$, we can form a bootstrap estimator of the variance:

$$\widehat{\text{Var}}_{\text{boot}}(\hat{R}_{\text{LURE}}) = \frac{1}{B-1} \sum_{b=1}^{B} (\hat{R}_{\text{boot}}^b - \bar{R}_{\text{boot}})^2, \quad \bar{R}_{\text{boot}} = \frac{1}{B} \sum_{b=1}^{B} \hat{R}_{\text{boot}}^b. \tag{6}$$

It is important to note that $\hat{R}_{\text{boot}}$ is not a standard bootstrap estimator because the reweighted losses, $L_j$, are not independent and identically distributed (they are the result of active data acquisition), and the bootstrap resampling process does not capture the dependencies between the losses. As a result, the estimator lacks theoretical convergence guarantees. Nevertheless we find it to be reliably accurate in our experiments (Section 5.9), suggesting it could be useful in practice.

## 5 Experiments

We now seek to empirically assess our proposed approach for active testing of LLMs (Section 3), as well as our proposed risk-error-estimation method (Section 4). We provide implementation details in Appendix B and code at `github.com/gabrielleberrada/scaling-up-active-testing`.

### 5.1 Setup

We use five text-classification datasets: Stanford Sentiment Treebank 2 (SST-2; Socher et al, 2013), Subjectivity (Subj; Pang & Lee, 2004), Financial Phrase Bank (FPB; Malo et al, 2013), Hate Speech (HS; De Gibert et al, 2018) and Massive Multitask Language Understanding (MMLU; Hendrycks et al, 2021). We mainly focus on the first four, which were previously used by Kossen et al (2024), then use MMLU to explore a problem that is more challenging for the models used in our experiments.

The models we use span three model families and have parameter counts ranging from 1 to 70 billion. In our main experiments we use the 7B and 70B versions of Llama 2 (Touvron et al, 2023) along with Gemma3-4B (Kamath et al, 2025); in additional experiments we use Phi-2 (Abdin et al, 2023) and Gemma3-1B. We use these models with either zero- or few-shot prompting, with both including

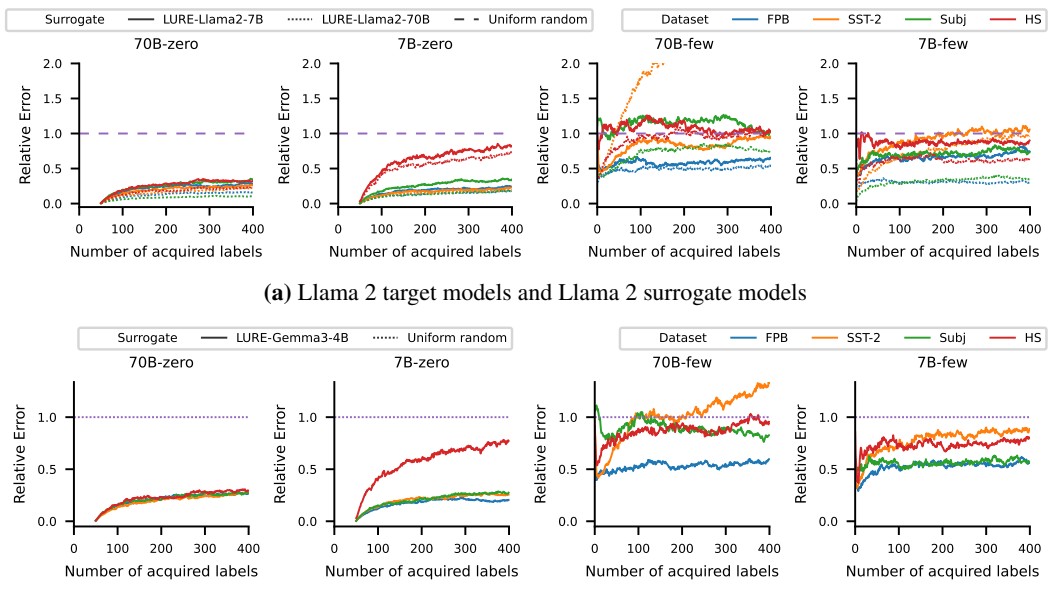

**(a)** Llama 2 target models and Llama 2 surrogate models

**(b)** Llama 2 target models and Gemma3-4B surrogate model

**Figure 2** Cheap surrogate models support effective data acquisition for active testing. We compare uniform-random testing to active testing (LURE) across four datasets. To guide active data acquisition we use a surrogate model that we train using a single step of in-context learning and then keep fixed. This stripped-back surrogate-model training, along with the use of small surrogate models relative to the target models, allows us to drastically reduce the computational cost of active testing while achieving strong performance.

appropriate task prompting such as "Classify the sentiment of the following sentence as positive or negative". We denote our four Llama 2 model configurations as 7B-zero, 70B-zero, 7B-few and 70B-few, and our two Gemma 3 model configurations as Gemma3-4B-zero and Gemma3-4B-few.

We compare active testing to uniform-random testing, a standard approach to model evaluation. Acquisition methods from the active-learning literature (Settles, 2012) are not tailored for risk estimation (Kossen et al, 2021) and so would have limited utility as baseline methods here.

We use a logarithmic loss, as discussed in Sections 2 and 3. Like in past work (Kossen et al, 2021, 2022), we measure the performance of a testing method using the median squared error of its risk estimate across random-number seeds, where the ground-truth risk is computed using a held-out test set. Alongside plots of the absolute value of this risk-estimation error, we present plots of *relative error*, namely the active risk-estimation error divided by the uniform-random risk-estimation error. Visualising performance on this relative scale lets us see the benefits of active testing more clearly. When we use a few-shot surrogate model with $n$ labelled in-context examples to evaluate a zero-shot target model, we ensure a fair comparison to uniform-random testing by comparing our results for $k$ acquired labels to uniform-random results for $n + k$ acquired labels.

## 5.2 Fixed surrogate models trained by in-context learning support effective data acquisition

First we investigate whether our switch from gradient-based training of the surrogate model to a single step of in-context learning still allows accurate risk estimation. We do this by running data acquisition across four datasets, four target models and three testing methods. In particular we compare uniform-random testing against two versions of active testing, LURE-7B and LURE-70B, differing only in the surrogate model used (7B-few vs 70B-few).

We find that active testing performs very well relative to the standard uniform-random testing in the vast majority of cases (Figure 2a), with risk-estimation error lowered by 32% on average (median over all the experimental variables listed above, as well as over possible label budgets between 1 and 400). This is an exciting result: a practically straightforward technique allows us to improve on standard practice for LLM evaluation with minimal computational overhead.

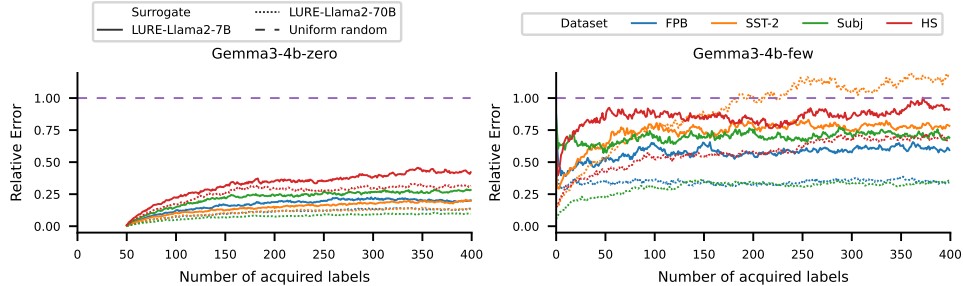

**Figure 3**   Older Llama 2 models are useful surrogates for active testing of newer Gemma 3 target models.

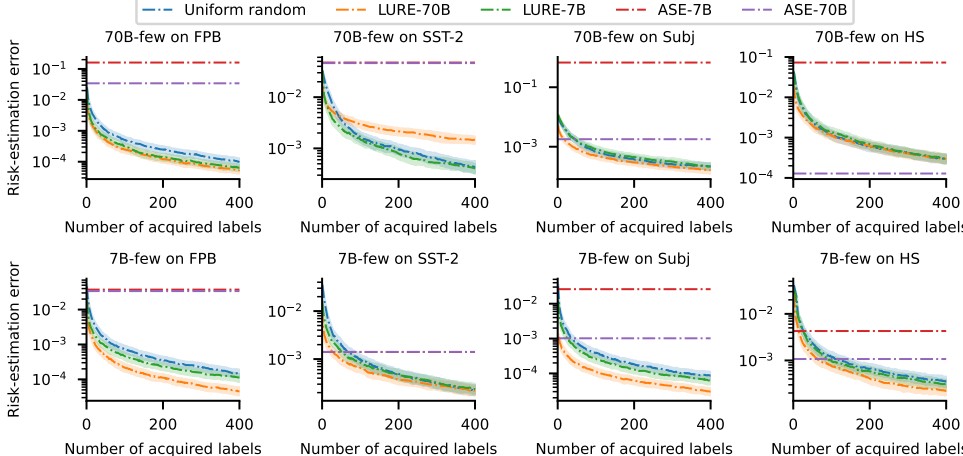

**Figure 4**   Our sampling-based active testing (LURE) approach works more reliably than interpolation-based active testing (ASE) when using cheaply constructed surrogate models. In ASE the surrogate model not only guides data acquisition but also much more directly affects the risk estimate: the labels used to compute the expected loss of the target model are simulated from the surrogate model.

Contrasting with the overall positive result in Figure 2a, LURE-70B performs poorly in evaluating the 70B-few model on the SST-2 dataset. We explore this failure case in Section 5.8 and show it is due to a large number of incorrect labels in the dataset. In Appendix C.5 we present an additional experiment that compares dynamic surrogate models to fixed ones: for the scenarios we study, dynamic surrogate models provide no accuracy benefits but incur prohibitive computational costs.

### 5.3 Small surrogate models can be used to evaluate larger target models

A key additional result to highlight in Figure 2a is that LURE-7B works well not just for evaluating a target model of the same size, but also for evaluating a target model ten times larger. In fact, we show that we can reduce surrogate-model costs even further by using Gemma3-4B (Figure 2b) and Phi-2 (Figure 10a) to evaluate Llama 2 models, reducing computational costs by using even smaller surrogates. Meanwhile, older Llama 2 models effectively evaluate newer Gemma 3 models (Figure 3), which perform strongly on these datasets (Tables 2 and 3). As discussed in Section 3.2, this ability to use a small surrogate model—combined with the fact that fixing the surrogate model allows us to only compute one forward pass on the pool set—is a useful way of reducing costs.

### 5.4 Using stripped-back surrogate models favours sampling-based active testing

Next we evaluate our claim in Section 3.1 that sampling-based active testing is preferable over interpolation-based active testing given that the surrogate model is fixed. We do this by comparing our approach with ASE in evaluating the 7B-few and 70B-few models. Our results show ASE produces strong results on the HS dataset with the 70B-few model, but otherwise significantly underperforms (Figure 4). This is consistent with our explanation that ASE is more sensitive to the quality of the

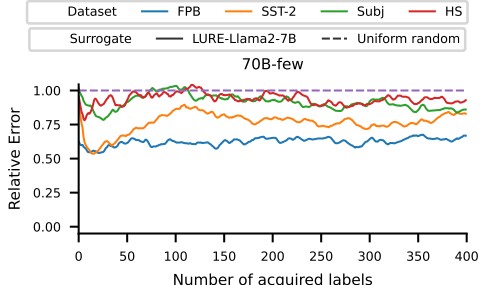

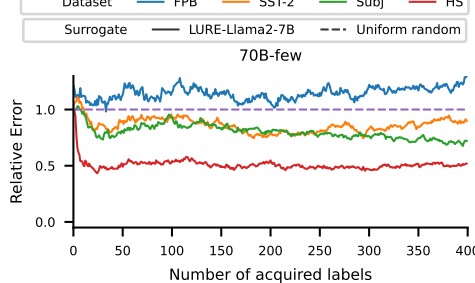

**Figure 5** Using the surrogate model to approximate the target model during data acquisition (here this means acquiring data based on the predictive entropy of the surrogate model) reduces the need for target-model predictions while still performing well.

**Figure 6** Active testing with a label-aware acquisition function (here the negative log likelihood of the surrogate model) enables dataset curation (selecting a subset of already-labelled test data) tailored to a target model of interest, helping reduce computational costs.

surrogate model, and that computational constraints on the surrogate model make it less practically useful. Note that the ASE risk estimator is constant if the surrogate model is fixed, as it is here.

## 5.5 Good data acquisition is possible without making predictions with the target model

Now we explore our suggestion in Section 3.3 that the surrogate model can be used to approximate the target model, such that we do not require target-model predictions during data acquisition. For the logarithmic loss we are using, this equates to using the predictive entropy of the surrogate model as the acquisition function. We assess this approach as applied to evaluating the 70B-few model with LURE-7B, and find it is surprisingly effective (Figure 5). This is promising: it suggests all the cost-reduction measures proposed in Section 3 are compatible with strong active-testing performance.

## 5.6 Active testing for dataset curation works, suggesting scope to reduce computational costs

Here we consider the alternative problem setting discussed in Section 3.4, where the inputs in the pool set are already labelled but it is too expensive to compute target-model predictions on all of them, with a possible solution being to curate a subset using a cheap surrogate model. More concretely we run active testing using the negative log likelihood as our label-aware acquisition function (Equation 4).

We find our approach improves over uniform-random subsampling on three datasets out of four (SST-2, Subj and HS) and performs slightly worse on FPB (Figure 6). These overall-positive results—along with those for the predictive-entropy function we considered in Section 5.5—suggest that active testing could be a useful tool for dataset curation, reducing computational costs when using existing evaluation datasets. The practical benefits of this could be significant: Liang et al (2023) required almost 20,000 hours of (Nvidia A100) GPU hours to evaluate 30 models on their HELM benchmark.

## 5.7 The benefit of active testing tends to hold in more challenging scenarios

Next we use the MMLU dataset to assess the robustness of active testing to an increase in task difficulty, which corresponds to worse surrogate-model performance (Tables 2 and 3) and thus poses a challenge for effective data acquisition. Our results show active testing continuing to outperform uniform-random testing for most combinations of surrogate and target models, with the only failure occurring when using 70B-few for both the surrogate and target models (Figure 7).

## 5.8 Incorrect labels can cause problems for active testing

Now we investigate the failure of LURE-70B to effectively evaluate the 70B-few model on the SST-2 dataset (Section 5.2). We construct two modified versions of the SST-2 pool set: one filtered to remove inputs for which 70B-few's negative log likelihood (NLL) is greater than 5, and one filtered with a NLL threshold of 3. Then we re-run active testing with these modified pool sets.

We show that the failure case is reduced by the weaker filtering and completely resolved by the stronger filtering (Figure 8). To understand why, we inspect the filtered-out data and find it is often

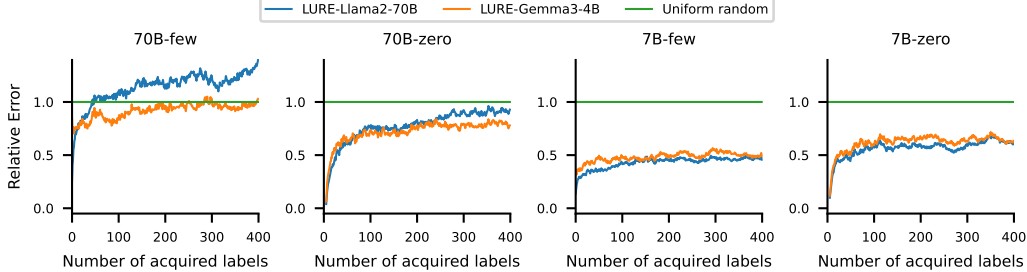

**Figure 7** Active testing outperforms uniform-random testing in most cases on the harder MMLU dataset.

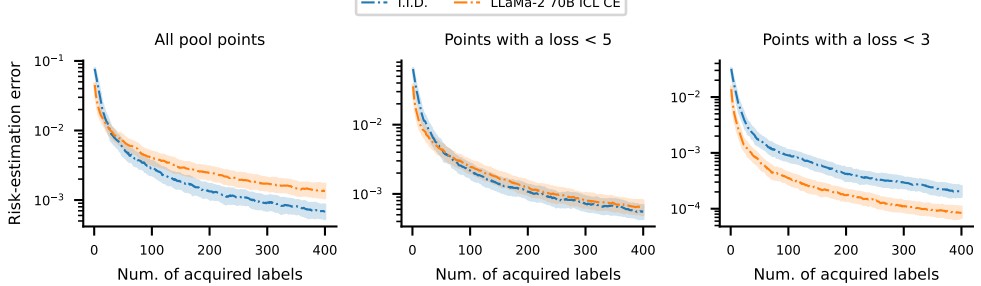

**Figure 8** The failure of LURE-70B in efficiently evaluating the 70B-few model on the SST-2 dataset (left) is resolved by filtering incorrectly labelled datapoints from the pool set (middle, right). Underlying this effect is the presence of incorrect labels in the original dataset: we find that inputs with high loss are often mislabelled.

mislabelled. We find 75% of the 45 examples with NLL greater than 5 are mislabelled, and 67% of the 136 examples with NLL greater than 3. This is a useful demonstration of how active testing can fail when the source of labels is unreliable. Interestingly it can also be seen as a case for using less powerful surrogate models to guide data acquisition: LURE-7B worked well in the exact case where LURE-70B failed. It is also worth noting that in a dataset-curation setting (Section 5.6) we have access to all the test labels and therefore have the option to directly filter out high-NLL examples.

### 5.9 Bootstrap estimation of risk-estimation error provides an accurate performance indicator

Having assessed our proposed active-testing method, we turn to our method for estimating the mean squared error of the LURE risk estimator (Section 4). We run active testing $T = 100$ times, generating 100 sequences of acquired indices, $(i_{1:M})_{j=1}^{100}$. Then for $K \in \{1, 2, \ldots, M\}$ we perform four steps:

1. Compute a ground-truth mean-squared error, $\mathrm{MSE}_K$, using Equation 5 with $q(i_{1:K})$ defined as a uniform distribution over the $K$ index sequences acquired so far.

2. Compute a bootstrap estimate of the MSE, $\widehat{\mathrm{MSE}}_K$, using Equation 6 with $B = 1000$ estimates.

3. Compute the MSE-estimation error, $(\mathrm{MSE}_K - \widehat{\mathrm{MSE}}_K)^2$.

4. Compute an approximate confidence interval, $\widehat{\mathrm{MSE}}_K \pm 2\hat{\sigma}$, using estimated standard deviation $\hat{\sigma}$.

Our results show MSE-estimation error converging to low values for most runs, after an initial period ($K < 100$) of relatively high error (Figure 9). In addition to this, our confidence intervals provide high levels of coverage: for a strong majority of runs we see coverage probabilities of around 94% for $K \geq 100$, suggesting a practitioner can expect their computed confidence interval to contain the true mean squared error approximately 94% of the time. While there are cases where MSE estimation fails, the overall performance is promising, suggesting our proposed method could be a useful practical performance indicator, helping justify real-world deployment of active testing.

## 6 Related work

The sampling-based approach to active testing that our approach builds on was proposed by Kossen et al (2021), using risk estimators from Farquhar et al (2021). Earlier work on active testing proposed

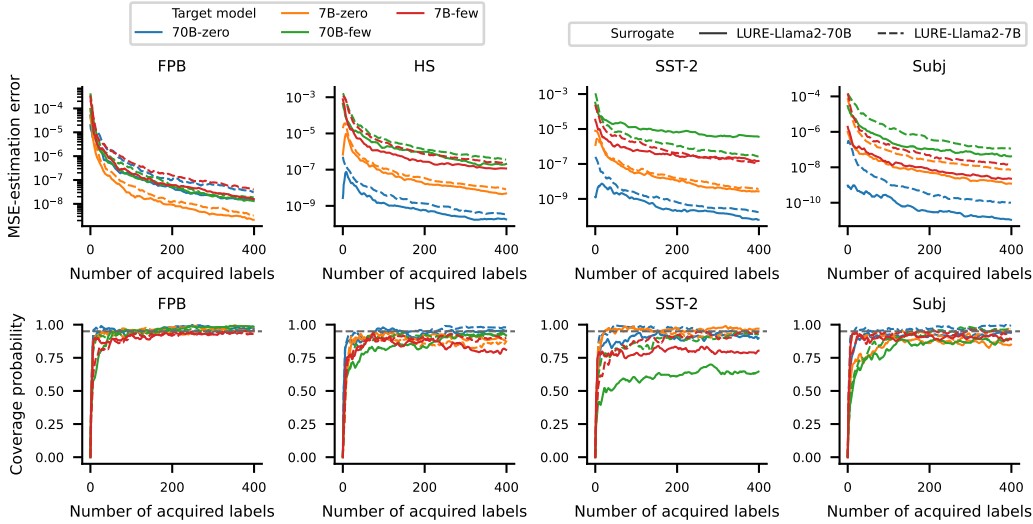

**Figure 9**   Our bootstrap estimator of the active-testing risk-estimation error provides a useful performance indicator. Coverage probability measures how often a confidence interval contains the true risk-estimation error.

alternative methods. Bennett & Carvalho (2010), Ji et al (2021), Katariya et al (2012) and Kumar & Raj (2018) explored stratification-based techniques, such as splitting the pool set into strata (based on a measure of model confidence) and sampling uniformly within each stratum. Sawade et al (2010) and Yilmaz et al (2021) proposed importance-sampling and Poisson-sampling methods related to that of Kossen et al (2021) but without the latter's innovations of adaptive, surrogate-based data acquisition and risk estimation that accounts for sampling without replacement. Nguyen et al (2018) studied a special case of active testing focused on human vetting of noisy labels.

Recent years have seen various extensions of this earlier active-testing work. Huang et al (2025) introduced a clustering-based approach that relies on making predictions with the target model on the pool set, which we argued is a barrier to scaling up. Su et al (2024) proposed a technique tailored to "dense" recognition tasks in computer vision (e.g., image segmentation and object detection). Yu et al (2024) incorporated active testing into the process of training a model. Ashury-Tahan et al (2024) and Hara et al (2024) proposed methods for model selection.

The idea of reducing model-evaluation costs by using carefully selected test datasets has been highlighted a number of times in recent work. Maynez et al (2023) found that the datasets they studied could be substantially reduced in size, through uniform-random subsampling, while maintaining stable rankings of the models they were comparing. Polo et al (2024), Saranathan et al (2024) and Vivek et al (2024) demonstrated more sophisticated methods for achieving the same goal. The goal in these studies was to take an existing test dataset (with labels for all inputs) and reduce it in a way that supports comparisons between different models. This prior work is therefore complementary to our contribution: we primarily study the problem of acquiring new data with which to evaluate a model of interest; and we use a method that tailors data acquisition to that given model.

# 7   Conclusion

We have argued that a significant barrier to principled active testing of large models is the computational cost of deciding which labels to acquire. To address this we have identified key contributors to the cost—training the surrogate model, and making predictions with the surrogate and target models—and have proposed straightforward cost-saving measures. We have shown that these measures are compatible with effective active testing, producing low-error risk estimates for large language models. On top of this we have demonstrated an estimator of risk-estimation accuracy that can provide an on-the-fly indication of how well active testing is working on a given practical problem. Overall we believe this represents substantial progress in more efficiently evaluating LLMs.

## Acknowledgements

Freddie Bickford Smith was supported by the EPSRC CDT in Autonomous Intelligent Machines and Systems (EP/L015897/1). Tom Rainforth was supported by EPSRC grant EP/Y037200/1.

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

## A  Impact statement

This paper presents work whose goal is to generally advance label- and compute-efficient model evaluation. As this is not limited to a particular area of application, it is difficult to evaluate the impact of this work. Generally speaking, we believe our work could play a role in reducing the computational costs, and thus the $CO_2$ footprint, associated with model evaluation. As is often the case with research, we would caution against careless use of our method in real-world applications.

## B  Experiment details

### B.1  Datasets

Our experiments are based on five datasets: Stanford Sentiment Treebank 2 (SST2; Socher et al, 2013; unknown license), Subjectivity (Subj; Pang & Lee, 2004; Creative Commons Attribution 4.0 International License), Financial Phrase-bank (FPB; Malo et al, 2013; Creative Commons Attribution Non Commercial Share Alike 3.0 Unported License), Hate Speech (HS; De Gibert et al, 2018; Creative Commons Attribution-Share Alike 3.0 Spain License) and Massive Multitask Language Understanding (MMLU; Hendrycks et al, 2021; MIT license).

Each method, including uniform-random testing, acquires data from the same pool set (a set of candidate data for acquiring, with labels hidden until they are acquired), although its size may vary between experiments. We compare risk estimates against a ground-truth risk, which is computed on a separate test dataset in all cases except for FPB, where we use the pool set due to the small size of the original dataset. Table 1 summarizes sizes of the pool and test sets for each dataset.

|          | SST-2  | Subj  | FPB   | HS    | MMLU  |
|----------|--------|-------|-------|-------|-------|
| Pool set | 10,000 | 6,000 | 2,200 | 6,000 | 7,000 |
| Test set | 10,000 | 4,000 | –     | 4,000 | 7,000 |

**Table 1**  Pool-set and test-set sizes for each dataset. Sets are randomly sampled so that they are disjoint.

**Stanford Sentiment Treebank 2 (SST-2)**  The Stanford Sentiment Treebank 2 dataset consists of 69,000 human-annotated sentences extracted from movie reviews. Each sentence is labeled as either "positive" or "negative", providing a benchmark for assessing sentiment-analysis capabilities. We use the predefined split of the dataset, concentrating on the training set, which contains 56% positive and 44% negative labels. We randomly select two subsets of 10,000 sentences each from the training set to form the pool and test sets. The instruction used for this dataset is

```
Classify the sentiment of the following sentence as "positive" or "negative".
Respond with "positive" or "negative".\n.
```

The string `"Answer"` is replaced by `"Label"` for this dataset only.

**Subjectivity (Subj)**  The Subjectivity dataset is composed of 5,000 subjective sentences from movie reviews and 5,000 objective sentences from plot summaries. This dataset is used to determine sentiment polarity. The task is to classify each sentence as either "subjective" or "objective". This dataset is balanced, with 49.5% of objective and 50.5% of subjective sentences. We randomly split the 10,000-sentence train set into a 6,000-sentence pool set and a 4,000-sentence test set. Examples for in-context-learning are selected from the separate 2,000-sentence test set. The instruction used for this dataset is

```
Is the following sentence 'objective' or 'subjective'. Respond with 'objective' or
'subjective'.\n.
```

**Financial Phrase-bank (FPB)**  The Financial Phrase-bank dataset consists of 4,800 English financial news articles that were classified as "positive", "neutral" and "negative" by human experts. We select the training subset of 2,200 sentences for which all 16 annotators agreed. This set contains 13% of negative, 25% of positive and 61% of neutral sentences. Due to its reduced size, the whole set

is used as the pool set, and the approximation of the exact loss is the loss over the whole set. The instruction used for this dataset is

```
Classify the sentiment of the following sentence as "negative", "neutral" or
"positive".Respond with "negative", "neutral" or "positive".\n.
```

**Hate Speech (HS)**  The Hate Speech dataset is composed of 9,900 sentences extracted from Stormfront, a white-supremacist forum, from which we select the 9,600 sentences which are classified as "hate" or "no hate" to obtain a binary-classification task. The dataset is highly unbalanced, with 11% classified as hate speech and 89% as non-hate speech. We randomly split this dataset into a pool set of size ∼6,000 and a test set of size ∼4,000. The instruction used for this dataset is

```
Does the sentence contain hate speech?Respond with "yes" or "no".\n.
```

**Massive Multitask Language Understanding (MMLU)**  The Massive Multitask Language Understanding dataset is composed of 116,000 sentences across 57 tasks asssing world knowledge and problem-solving. Each question is multiple-choice, with four choices available. We randomly split this dataset into a pool set of size 7,000 and a test set of size 7,000. The instruction used for this dataset is

```
Answer the question with the correct letter.Respond with only 'A', 'B', 'C' or
'D'.\n.
```

## B.2  Models

We use the 7B and 70B models from the Llama 2 family (Touvron et al, 2023; Llama 2 Community License). We use 8 bit-quantisation for the 7B model and half-precision floating-point numbers for the 70B model; Kossen et al (2024) showed that these approximations do not significantly affect performance. We additionally use Gemma-3 4B (Kamath et al, 2025; Gemma License).

## B.3  Prompt formatting

To evaluate a model on a classification task, we follow the formatting guidelines from Kossen et al (2024) in how we generate the input sentence. We begin with a dataset-specific instruction, such as

```
Classify the sentiment of the following sentence as "[label1]" or "[label2]".
Respond with "[label1]" or "[label2]".\n.
```

Then we introduce the sentence to be classified and ask for the corresponding label. The prompt is formatted as

```
Sentence:'[sentence]' \nAnswer:.
```

## B.4  In-context examples

We randomly select in-context examples such that each class is represented in proportion to its actual proportion in the dataset, which we found improved the model's accuracy. These examples are included in the input between the instruction and the prompt. Each input example is formatted as

```
Sentence:'[sentence]' \nAnswer:[label]\n\n.
```

All few-shot models are given 50 in-context examples for all datasets, except for the MMLU dataset, for which models receive only 5 examples due to token limit. Examples are ordered randomly once and fixed for all evaluations, ensuring that all models receive the same context. The set of in-context examples is therefore fixed beforehand and is common to both the target model and the surrogate model, for label efficiency and fair comparison.

## B.5  Generating and processing model outputs

To obtain deterministic, reproducible token generation, we set the maximum number of tokens to 1, do not set a top-$k$ nor top-$p$ value and output the logits directly, from which we compute probabilities.

For each input, the model outputs a logit value for each token in the vocabulary. We select the logits corresponding to the labels of the dataset and apply the softmax function to obtain a final probability distribution over the possible labels. Any logit that does not correspond to one of the labels is ignored.

In cases where a model represents a task label with multiple tokens (Kossen et al, 2024), we select only the relevant token that corresponds to the core of the word. For instance, the Llama 2 tokeniser encodes the word "objective" as [12091] but "subjective" as [4967, 573], which are tokens for "subject" and "ive". In this case we select [12091] for "objective" and [4967] for "subjective".

### B.6 Data acquisition

Following Kossen et al (2021), we clip values of the acquisition function to ensure no zero-mass inputs and thus guarantee that the LURE is unbiased. All acquisition probabilities below $\alpha = 0.1$ times the probability corresponding to uniform-random acquisition are brought up to this limit value.

### B.7 Computational resources

Our experiments are designed to be computationally efficient and therefore do not require substantial resources. Generating model outputs over the pool set is the main computational cost, typically requiring two GPUs for Llama 2 70B and one GPU for all other models used in this work. Aside from this, the active-testing procedure itself can be run efficiently on a small number of CPUs.

## C Additional results

### C.1 Model performance

Here we present accuracy and loss values for Llama2-7B, Llama2-70B and Gemma3-4B on the five datasets we study. Results for Llama 2 here match those published in Kossen et al (2024).

| Dataset | L-7B-zero | L-7B-few | L-70B-zero | L-70B-few | G-4B-zero | G-4B-few |
|---------|-----------|----------|------------|-----------|-----------|----------|
| FPB | 66.75 | 90.60 | 26.25 | 94.15 | 29.70 | 91.00 |
| SST-2 | 63.96 | 92.19 | 76.80 | 93.60 | 57.79 | 91.68 |
| Subj | 49.38 | 89.72 | 54.85 | 96.12 | 50.40 | 93.73 |
| HS | 11.38 | 89.28 | 84.18 | 90.23 | 42.87 | 89.42 |
| MMLU | 35.41 | 42.16 | 60.80 | 65.34 | 52.61 | 57.83 |

**Table 2**  Accuracy (%) of Llama 2 (L) 7B and 70B models and Gemma3-4B (G) model evaluated on pool sets.

| Dataset | L-7B-zero | L-7B-few | L-70B-zero | L-70B-few | G-4B-zero | G-4B-few |
|---------|-----------|----------|------------|-----------|-----------|----------|
| FPB | 0.7847 | 0.3084 | 1.3123 | 0.2152 | 1.4695 | 0.2701 |
| SST-2 | 0.5639 | 0.2050 | 0.5808 | 0.1828 | 0.6234 | 0.2107 |
| Subj | 0.7326 | 0.3278 | 0.5873 | 0.1235 | 0.7101 | 0.2741 |
| HS | 0.9129 | 0.2979 | 0.5386 | 0.2166 | 0.7245 | 0.2678 |
| MMLU | 1.3468 | 1.2568 | 0.9217 | 0.8384 | 1.0920 | 0.9683 |

**Table 3**  Loss of Llama 2 (L) 7B and 70B models and Gemma3-4B (G) model evaluated on pool sets.

### C.2 Failure mode on Subjectivity

Here we build on our failure analysis in Section 5.8 (where we investigated why LURE-70B failed for the 70B-few target model on the SST-2 dataset), now focusing on the underperformance of LURE-7B for the 70B-few target model on the Subj dataset. We find that filtering out high-loss examples does not improve performance like it did in the SST-2 case, but we do find some insight from another

| Dataset | 70B (T) & 70B (S) | 70B (T) & 7B (S) | 7B (T) & 70B (S) | 7B (T) & 7B (S) |
|---|---|---|---|---|
| FPB | 0.628 | 0.392 | 0.707 | 0.588 |
| SST-2 | **0.285** | 0.224 | 0.527 | 0.454 |
| Subj | 0.364 | **− 0.025** | 0.802 | 0.652 |
| HS | 0.461 | 0.146 | 0.555 | 0.384 |

**Table 4** Pearson's correlation coefficient between the cross-entropy of the surrogate (S) and target (T) models' predictions and the negative log likelihood of the target model's predictions. Low or negative correlation indicates poor alignment between surrogate-based acquisition and optimal acquisition. Failure cases are in bold.

line of investigation. In particular we compute the Pearson correlation coefficient between the cross-entropy-based acquisition scores and the optimal acquisition function (negative log likelihood of the target model's predictions), with this correlation quantifying how well surrogate-based acquisition approximates optimal acquisition. Notably this is the only setting where the correlation is close to zero and is negative. This indicates that the surrogate model provides no meaningful guidance, helping to explain why active testing performs similarly to uniform-random testing in this case.

### C.3    Smaller surrogate models on the four core datasets

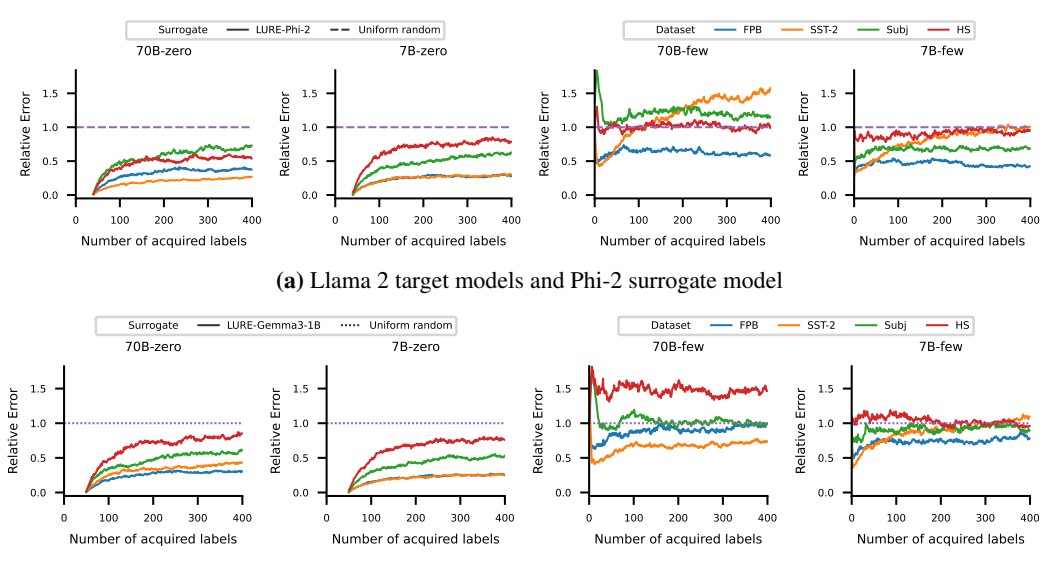

**(a)** Llama 2 target models and Phi-2 surrogate model

**(b)** Llama 2 target models and Gemma3-1B surrogate model

**Figure 10** Even very small surrogate models can support effective active testing in simple cases, although they struggle with evaluating our best-performing target model, 70B-few.

Here we explore two surrogate models that are smaller than those used in Section 5: Phi-2 2.7B (Abdin et al, 2023; MIT License) and Gemma3-1B (Kamath et al, 2025; Gemma License). We see both surrogate models producing strong performance in evaluating 70B-zero and 7B-zero, and more mixed performance in evaluating the two stronger target models, 70B-few and 7B-few (Figure 10).

### C.4    Smaller surrogate model on MMLU

Here we revisit Gemma3-1B as a surrogate model, now on the harder MMLU dataset. We see active testing failing to outperform uniform-random testing (Figure 11). This can be explained by how poorly Gemma3-1B performs on MMLU: it achieves approximately 25% accuracy, the base accuracy produced by uniform-random prediction. A weak surrogate model can thus undermine active testing.

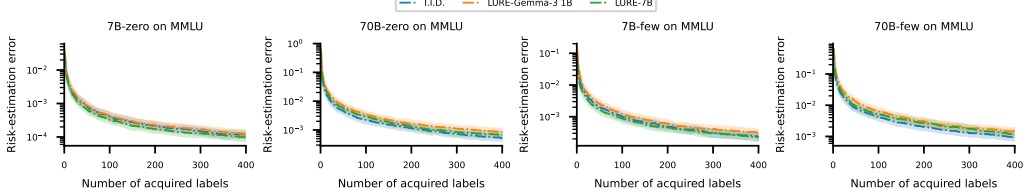

**Figure 11** A sufficiently weak surrogate model can cause active testing to consistently fail on MMLU.

## C.5 Dynamic-surrogate active testing

Here we explore the use of dynamic surrogate models, in contrast with the fixed surrogate models we use in our approach. In the dynamic approach the surrogate model's context incorporates newly acquired labels, leading to recomputed predictions, and thus novel acquisition probabilities at each sampling step. Our experiments focus on few-shot Llama2-7B and Llama2-70B target models with dynamic and fixed Llama2-7B surrogate models. We start with 10 in-context examples and run 40 steps of data acquisition. We run for 50 random-number seeds for dynamic-surrogate active testing, and for 3,000 seeds for fixed-surrogate active testing and uniform-random active testing.

Despite being much more computationally expensive, dynamic-surrogate active testing shows no consistent improvement in risk estimation over fixed-surrogate active testing (Tables 5 and 6). Thus, while dynamic surrogate models might be useful elsewhere, the results we see for our selection of datasets support our design choice of fixing the surrogate model: it improves over uniform-random testing while maintaining a low computational cost and thus readily scaling to LLM evaluations.

| Dataset | Testing method | Step 10 | Step 20 | Step 30 | Step 40 |
|---------|----------------|---------|---------|---------|---------|
| SST-2 | Uniform random | 91.41 (87.04, 95.86) | 45.49 (43.53, 47.52) | 29.25 (28.00, 30.51) | 20.66 (19.82, 21.52) |
| | LURE-7B dynamic | 63.25 (36.23, 96.67) | 22.79 (14.56, 32.91) | 13.52 (8.93, 18.80) | 8.62 (5.82, 11.88) |
| | LURE-7B fixed | 42.71 (40.32, 45.11) | 19.99 (19.07, 20.96) | 12.42 (11.87, 13.00) | 8.74 (8.38, 9.11) |
| FPB | Uniform random | 135.75 (130.00, 141.72) | 65.14 (62.37, 68.05) | 40.06 (38.41, 41.76) | 27.99 (26.84, 29.16) |
| | LURE-7B dynamic | 65.73 (45.79, 86.86) | 36.40 (25.31, 49.18) | 18.33 (11.82, 25.79) | 11.53 (7.37, 16.43) |
| | LURE-7B fixed | 77.54 (73.39, 81.94) | 36.88 (35.22, 38.61) | 22.97 (21.99, 23.95) | 16.26 (15.56, 16.96) |
| HS | Uniform random | 156.59 (152.02, 161.10) | 96.51 (93.78, 99.32) | 77.32 (75.14, 79.54) | 66.84 (64.93, 68.76) |
| | LURE-7B dynamic | 121.45 (100.99, 142.89) | 75.61 (60.04, 91.73) | 52.12 (40.48, 64.84) | 43.83 (35.35, 52.63) |
| | LURE-7B fixed | 144.44 (139.38, 149.87) | 89.65 (87.31, 91.98) | 72.62 (70.73, 74.47) | 63.67 (62.07, 65.31) |
| Subj | Uniform random | 46.97 (44.97, 49.00) | 21.79 (20.89, 22.70) | 13.76 (13.19, 14.36) | 10.23 (9.80, 10.66) |
| | LURE-7B dynamic | 71.76 (50.93, 95.54) | 35.25 (20.62, 53.32) | 36.75 (21.11, 55.19) | 30.94 (18.43, 46.09) |
| | LURE-7B fixed | 40.20 (38.39, 42.02) | 19.69 (18.89, 20.48) | 12.95 (12.42, 13.50) | 9.12 (8.73, 9.52) |

**Table 5** Mean squared error (multiplied by $10^4$; mean on first row; 5th and 95th percentile on second row) for uniform-random testing, dynamic-surrogate active testing and fixed-surrogate active testing of 7B-few.

| Dataset | Testing method | Step 10 | Step 20 | Step 30 | Step 40 |
|---------|----------------|---------|---------|---------|---------|
| SST-2 | Uniform random | 81.75 (77.77, 85.85) | 40.04 (38.24, 41.87) | 25.01 (23.98, 26.06) | 17.92 (17.19, 18.66) |
| | LURE-7B dynamic | 28.76 (22.13, 35.70) | 17.21 (11.42, 23.89) | 12.62 (8.83, 16.74) | 8.57 (6.37, 11.07) |
| | LURE-7B fixed | 40.09 (38.28, 41.98) | 19.11 (18.35, 19.89) | 12.26 (11.80, 12.73) | 8.44 (8.12, 8.76) |
| FPB | Uniform random | 173.90 (164.57, 183.50) | 80.36 (76.55, 84.22) | 50.14 (47.77, 52.54) | 35.07 (33.40, 36.83) |
| | LURE-7B dynamic | 114.11 (84.88, 144.11) | 34.89 (25.00, 46.22) | 19.71 (13.20, 27.05) | 11.10 (7.68, 14.95) |
| | LURE-7B fixed | 150.45 (140.12, 161.06) | 70.59 (66.56, 74.69) | 45.85 (43.59, 48.21) | 33.25 (31.64, 34.85) |
| HS | Uniform random | 98.22 (94.33, 102.28) | 48.73 (46.95, 50.54) | 33.04 (31.84, 34.29) | 25.27 (24.36, 26.20) |
| | LURE-7B dynamic | 88.73 (64.35, 116.85) | 41.15 (31.37, 51.26) | 23.93 (18.57, 29.54) | 16.18 (12.40, 20.03) |
| | LURE-7B fixed | 74.57 (71.50, 77.75) | 39.31 (37.78, 40.87) | 26.87 (25.85, 27.89) | 19.93 (19.18, 20.69) |
| Subj | Uniform random | 109.53 (105.47, 113.54) | 66.96 (64.75, 69.11) | 51.41 (49.77, 53.10) | 42.97 (41.59, 44.36) |
| | LURE-7B dynamic | 209.81 (161.39, 260.62) | 134.48 (99.07, 172.06) | 72.61 (54.29, 92.06) | 64.43 (48.04, 82.24) |
| | LURE-7B fixed | 134.87 (129.79, 140.22) | 77.31 (74.56, 80.06) | 57.14 (55.20, 59.07) | 47.24 (45.64, 48.81) |

**Table 6** Mean squared error (multiplied by $10^4$; mean on first row; 5th and 95th percentile on second row) for uniform-random testing, dynamic-surrogate active testing and fixed-surrogate active testing of 70B-few.

## C.6 Risk-estimation values

In addition to the plots shown in Section 5, we present numerical values for the risk-estimation error of uniform-random testing and active testing (with the cross-entropy acquisition function) at acquisition steps 50, 100, 200, 300 and 400 in Tables 7 to 11.

| Target model | Testing method | Step 50 | Step 100 | Step 200 | Step 300 | Step 400 |
|---|---|---|---|---|---|---|
| 70B-zero | Uniform random | 24.3373 | 11.5663 | 5.1944 | 3.4854 | 2.4122 |
| | LURE-Llama2-7B | — | 8.0511 | 3.6772 | 1.7589 | 1.0463 |
| | LURE-Llama2-70B | — | **4.2225** | **2.0045** | **0.98390** | **0.64870** |
| | LURE-Gemma3-4B | — | 7.1132 | 3.5176 | 1.6571 | 1.0540 |
| 7B-zero | Uniform random | 9.4687 | 4.8302 | 2.1823 | 1.4524 | 1.0075 |
| | LURE-Llama2-7B | — | 2.5167 | 1.3094 | 0.59430 | 0.37380 |
| | LURE-Llama2-70B | — | 1.9294 | 0.96850 | 0.42460 | 0.27060 |
| | LURE-Gemma3-4B | — | **1.1421** | **0.5849** | **0.3681** | **0.2341** |
| 70B-few | Uniform random | 10.5749 | 5.16310 | 2.48680 | 1.51150 | 0.977800 |
| | LURE-Llama2-7B | 6.3984 | 3.2058 | 1.3719 | 0.89110 | 0.63880 |
| | LURE-Llama2-70B | 5.0757 | **2.5057** | **1.1921** | **0.76740** | **0.54300** |
| | LURE-Gemma3-4B | **5.0390** | 2.6267 | 1.3102 | 0.8344 | 0.5825 |
| 7B-few | Uniform random | 14.3272 | 7.2276 | 3.4608 | 2.3147 | 1.5246 |
| | LURE-Llama2-7B | 8.9252 | 4.5853 | 2.3314 | 1.5486 | 1.1345 |
| | LURE-Llama2-70B | **4.8174** | **2.1675** | **1.1186** | **0.65290** | **0.45770** |
| | LURE-Gemma3-4B | 6.5254 | 3.7684 | 1.9489 | 1.2371 | 0.8556 |

**Table 7** Risk-estimation error (multiplied by $10^4$; minimum per target model in bold) for the FPB dataset.

| Target model | Testing method | Step 50 | Step 100 | Step 200 | Step 300 | Step 400 |
|---|---|---|---|---|---|---|
| 70B-zero | Uniform random | 2.1154 | 1.0992 | 0.56060 | 0.34910 | 0.26150 |
| | LURE-Llama2-7B | — | 0.5386 | 0.3089 | 0.1551 | 0.1023 |
| | LURE-Llama2-70B | — | **0.4191** | **0.2308** | **0.1255** | **0.08190** |
| | LURE-Gemma3-4B | — | 0.5283 | 0.2790 | 0.1542 | 0.1042 |
| 7B-zero | Uniform random | 12.4455 | 5.49 | 2.9545 | 1.9694 | 1.362 |
| | LURE-Llama2-7B | — | 2.7160 | 1.4229 | 0.66190 | 0.48410 |
| | LURE-Llama2-70B | — | **2.3740** | **1.2437** | **0.60510** | **0.42530** |
| | LURE-Gemma3-4B | — | 3.5371 | 1.6843 | 0.8699 | 0.6062 |
| 70B-few | Uniform random | 36.6017 | 16.8825 | 9.4717 | 6.3474 | 4.3089 |
| | LURE-Llama2-7B | **24.8052** | **14.7678** | **7.9162** | **5.2615** | **4.0596** |
| | LURE-Llama2-70B | 40.8407 | 30.077 | 21.3892 | 17.1808 | 14.7939 |
| | LURE-Gemma3-4B | 26.3377 | 16.8812 | 9.4499 | 7.1826 | 5.7175 |
| 7B-few | Uniform random | 19.5177 | 9.5602 | 4.9347 | 3.1286 | 2.2497 |
| | LURE-Llama2-7B | 14.2747 | 8.3696 | 4.6265 | 3.2263 | 2.3764 |
| | LURE-Llama2-70B | **9.7216** | **6.4914** | **3.6886** | 2.8971 | 2.1722 |
| | LURE-Gemma3-4B | 13.3504 | 7.0426 | 3.9896 | **2.8028** | **1.9958** |

**Table 8** Risk-estimation error (multiplied by $10^4$; minimum per target model in bold) for the SST-2 dataset.

| Target model | Testing method | Step 50 | Step 100 | Step 200 | Step 300 | Step 400 |
|---|---|---|---|---|---|---|
| 70B-zero | Uniform random | 1.2104 | 0.64740 | 0.30890 | 0.21120 | 0.15430 |
| | LURE-Llama2-7B | — | 0.4688 | 0.2311 | 0.1113 | 0.07570 |
| | LURE-Llama2-70B | — | **0.139** | **0.0758** | **0.0373** | **0.0251** |
| | LURE-Gemma3-4B | — | 0.3809 | 0.1897 | 0.0887 | 0.0638 |
| 7B-zero | Uniform random | 9.7212 | 5.1028 | 2.4378 | 1.5612 | 1.1429 |
| | LURE-Llama2-7B | — | 3.4150 | 1.6835 | 0.87130 | 0.58730 |
| | LURE-Llama2-70B | — | **1.6958** | **0.89970** | **0.44150** | **0.29280** |
| | LURE-Gemma3-4B | — | 2.8896 | 1.4536 | 0.7144 | 0.4787 |
| 70B-few | Uniform random | 15.5609 | 6.6315 | 3.7465 | 2.5614 | 2.1347 |
| | LURE-Llama2-7B | 16.4075 | 8.2611 | 4.4942 | 3.1802 | 2.1987 |
| | LURE-Llama2-70B | **8.2473** | **5.0366** | **3.0052** | **2.1589** | **1.6293** |
| | LURE-Gemma3-4B | 12.615 | 6.7789 | 3.3058 | 2.2544 | 1.7645 |
| 7B-few | Uniform random | 7.4461 | 3.9287 | 1.8132 | 1.1308 | 0.87000 |
| | LURE-Llama2-7B | 5.2522 | 2.6268 | 1.2944 | 0.84500 | 0.63670 |
| | LURE-Llama2-70B | **1.8539** | **1.0921** | **0.62250** | **0.42070** | **0.30200** |
| | LURE-Gemma3-4B | 4.1246 | 2.1545 | 1.0256 | 0.6868 | 0.4941 |

**Table 9**  Risk-estimation error (multiplied by $10^4$; minimum per target model in bold) for the Subj dataset.

| Target model | Testing method | Step 50 | Step 100 | Step 200 | Step 300 | Step 400 |
|---|---|---|---|---|---|---|
| 70B-zero | Uniform random | 2.3826 | 1.1493 | 6.3630 | 0.39690 | 0.30590 |
| | LURE-Llama2-7B | — | 0.8497 | 0.4371 | 0.2247 | 0.1524 |
| | LURE-Llama2-70B | — | **0.5703** | **0.2996** | **0.1418** | **0.09900** |
| | LURE-Gemma3-4B | — | 0.7508 | 0.3700 | 0.1822 | 0.1330 |
| 7B-zero | Uniform random | 4.3455 | **2.3304** | **1.3545** | 0.96420 | 0.80470 |
| | LURE-Llama2-7B | — | 3.8653 | 1.9395 | 1.1326 | 0.86310 |
| | LURE-Llama2-70B | — | 3.0157 | 1.7693 | **0.94190** | **0.73000** |
| | LURE-Gemma3-4B | — | 3.5508 | 1.6981 | 0.9918 | 0.7766 |
| 70B-few | Uniform random | 24.5497 | 11.8593 | 5.8947 | 4.263 | 2.897 |
| | LURE-Llama2-7B | 26.4253 | 13.5746 | 6.9755 | 4.249 | 3.0014 |
| | LURE-Llama2-70B | 19.8693 | 11.2704 | 6.1626 | **3.983** | 2.9785 |
| | LURE-Gemma3-4B | **19.4658** | **10.4853** | **5.473** | 3.6247 | **2.767** |
| 7B-few | Uniform random | 23.7706 | 11.7404 | 6.6989 | 4.6215 | 3.4943 |
| | LURE-Llama2-7B | 19.6371 | 9.638 | 5.8037 | 4.0226 | 3.1452 |
| | LURE-Llama2-70B | **13.4897** | **7.5196** | **4.1388** | **2.7654** | **2.2691** |
| | LURE-Gemma3-4B | 16.6135 | 8.7585 | 4.8131 | 3.4505 | 2.8044 |

**Table 10**  Risk-estimation error (multiplied by $10^4$; minimum per target model in bold) for the HS dataset.

| Target model | Testing method | Step 50 | Step 100 | Step 200 | Step 300 | Step 400 |
|---|---|---|---|---|---|---|
| 70B-zero | Uniform random | 45.2305 | 22.0405 | 11.5532 | 7.2541 | 5.288 |
| | LURE-Llama2-7B | 57.137 | 28.4008 | 13.3008 | 8.4398 | 6.7655 |
| | LURE-Llama2-70B | **30.4079** | 17.3273 | 9.5205 | 6.6253 | 4.9939 |
| | LURE-Gemma3-4B | 30.894 | **16.3988** | **9.0229** | **5.6838** | **4.1935** |
| 7B-zero | Uniform random | 8.1557 | 4.1386 | 2.1924 | 1.4752 | 1.1799 |
| | LURE-Llama2-7B | 6.6835 | 3.2991 | 1.6763 | 1.2791 | 0.9767 |
| | LURE-Llama2-70B | **4.3678** | **2.6164** | **1.3063** | **0.9088** | **0.7330** |
| | LURE-Gemma3-4B | 4.9507 | 2.6543 | 1.4700 | 0.9569 | 0.7622 |
| 70B-few | Uniform random | 80.9096 | 41.8724 | 20.7615 | 12.5686 | 8.9944 |
| | LURE-Llama2-7B | 94.483 | 51.3035 | 27.2732 | 17.5476 | 11.713 |
| | LURE-Llama2-70B | 84.9646 | 45.9222 | 23.3469 | 15.7314 | 12.5486 |
| | LURE-Gemma3-4B | **71.571** | **34.8396** | **19.3527** | **12.9035** | **9.2558** |
| 7B-few | Uniform random | 20.5838 | 10.142 | 4.8003 | 3.0952 | 2.4424 |
| | LURE-Llama2-7B | 16.2533 | 8.686 | 4.5997 | 3.047 | 2.2239 |
| | LURE-Llama2-70B | **7.242** | **4.3427** | **2.1991** | **1.4639** | **1.1166** |
| | LURE-Gemma3-4B | 9.6443 | 4.9703 | 2.3827 | 1.694 | 1.2064 |

**Table 11** Risk-estimation error (multiplied by $10^4$; minimum per target model in bold) for the MMLU dataset.

