# OpenReview forum: "Scaling Up Active Testing to Large Language Models"
_NeurIPS.cc/2025/Conference — NeurIPS 2025 poster_

### Official Review · Reviewer_cFrJ · 2025-06-27

**Clarity:** 4
**Significance:** 4
**Originality:** 3
**Rating:** 5
**Confidence:** 3

**Summary:**

The authors extend the technique of active testing (Kossen et al., 2021) with active surrogate estimators (Kossen et al., 2022) to the domain of LLMs. The setting is that there is a large pool of test inputs, but acquiring gold labels for test inputs is expensive. Active testing is a label-efficient way to evaluate models in this setting by carefully selecting which test inputs to acquire gold labels for. At every label acquisition step, the next test input to label is sampled from a distribution based on an acquisition function that is based on the disagreement between the target model and a surrogate model that is trained on labelled test data acquired so far.

The authors’ contributions are showing that for computing the acquisition function: (1) the surrogate model can be a smaller LLM; (2) the surrogate model can be trained using in-context learning, thus eliminating the need for gradient-based training; (3) the surrogate model does not need to be retrained with new test labels at each step, thus requiring only a single inference pass over the test inputs; (4) the target model can also be replaced with the surrogate model, thus avoiding computing inferences with the target model. The authors validate their technique using experiments using Llama and Gemma family LLMs.

**Questions:**

- How were the datasets for the experiments selected?
- Did you consider running ablations for the individual components of your technique? I’m specifically curious about the ablation of adding previously acquired test labels into the in-context training of the surrogate models, rather than fixing the in-context training. I acknowledge that the ablations may have been prohibitively costly to run.

**Ethical Concerns:**

["NO or VERY MINOR ethics concerns only"]

**Final Justification:**

Overall, the main strengths of this paper are that the main proposed method is useful, practical, and well-evaluated. The main weakness is that its novelty is questionable. The use of in-context learning was obvious to myself and reviewer FX74. The use of an entropy-based acquisition method was obvious to reviewers x1Pz and FX74 - this was not obvious to me, but it is possible that this is due to my lack of familiarity with the field of active learning. Even so, I believe that the synthesis of these ideas is interesting, and that this paper makes valuable recommendations to the community on how to do cost-efficient LLM testing. This justifies my overall rating of Accept.

My other main concern was the lack of ablations. The authors have mostly addressed this by providing experimental results for some ablations (comparing dynamic surrogates with fixed surrogates).

**Limitations:**

The authors have adequately addressed the limitations of their work.

**Paper Formatting Concerns:**

None.

**Quality:**

4

**Strengths And Weaknesses:**

### Strengths

- The problem setting is very common in practical LLM evaluation settings. Gold labels are expensive because they require human labellers, and inference on the target model can be expensive for very large models. (Significance)
- The proposed technique is very practical and easy to use. (Quality, Significance)
- The paper is useful for introducing the active testing concept to the field of LLM evaluation, in which active testing is rarely used. (Significance)
- The prior work around active testing is well explained, and the proposed new technique is well-justified. (Clarity)
- The experiments are rigorous and produce convincing results. The investigation into the SST-2 dataset is explained well. (Quality)
- The idea of replacing the target model with the surrogate model is novel, and the insight that the acquisition function reduces to predictive entropy is elegant. (Originality)

### Weaknesses

- The authors conclude that the surrogate and target models do not have to be from the same family. However, the experiments only used two model families, so this result may not be very general. Additionally, the advantage of using a same-family surrogate model versus a different-family surrogate model was not quantified. This may be an area for future work. (Quality)
- The relevant importance of each of the individual components of the proposed technique is unclear, because there were no ablations for the individual components. (Quality)
- The graphs are too dense and hard to read. The graph lines make use of both color and style (dotted, dashed, solid), which is confusing. Consider splitting the current graphs further into multiple graphs. (Clarity)
- Some parts of the proposed technique (using in-context learning, using a smaller model) are somewhat obvious to LLM researchers. (Significance, Originality)
- The authors ran experiments using datasets that are relatively low-difficulty for recent LLMs in 2025, and did not include datasets requiring open-ended generation. Open-ended generation datasets would have been a more realistic reflection of the problem setting, because these datasets are expensive to acquire human labels for. (Quality)

---

> ### Author Rebuttal · Authors · 2025-07-30
>
> Thank you for your hard work and helpful feedback! We’re delighted that you appreciate the practical significance and rigorous experimental validation of our work. We hope our response below addresses your remaining questions.
>
> > Did you consider running ablations for the individual components of your technique? I’m specifically curious about the ablation of adding previously acquired test labels into the in-context training of the surrogate models, rather than fixing the in-context training. I acknowledge that the ablations may have been prohibitively costly to run. [...] The relevant importance of each of the individual components of the proposed technique is unclear, because there were no ablations for the individual components.
>
> Excellent suggestion! Following your feedback, we have conducted ablation studies comparing fixed versus dynamic surrogates, where the dynamic approach incorporates newly acquired labels into the surrogate’s context and recomputes predictions, and thus acquisition probabilities, at each sampling step.
> Our experiments evaluate the few-shot Llama-2 7B and 70B models with dynamic and fixed Llama-2 7B surrogates, starting with 10 in-context examples over 40 steps (50 runs each for dynamic surrogates, 3000 runs for fixed surrogates and uniform sampling). You can find a comparison of the mean relative error to random uniform sampling for acquisition with and without the dynamic surrogate below along with the 5th and 95th percentiles.
>
> In summary, we find that dynamic surrogates show no clear consistent improvements in risk estimation over fixed surrogates, while imposing prohibitive computational costs. The experiments with 50 repetitions each were over 1500 times more expensive for 40 steps than the frozen ones, whose cost remains fixed as the number of steps increases. Additionally, dynamic surrogates require unique predictions at each step, eliminating the ability to precompute and share predictions across multiple evaluations. While there may be interesting applications of dynamic surrogates, for our selection of tasks and models, the results support our design choice of fixed surrogates, which achieves the same accuracy benefits while maintaining the computational efficiency that makes this active testing method practical. We are happy to add these results to the revised version of the paper.
>
> ### Scaled mean squared error (x$10^{-4}$, 5th-95th percentile) for uniform random sampling, dynamic surrogate active testing, and fixed surrogate active testing on 7B-few and 70B-few across datasets
>
> | Acquisition step |                   | 10    | 20    | 30    | 40    |
> | -----------------|------------------ | ----- | ----- | ----- | ----- |
> | SST-2 7B-few  | Uniform random      | 91.41 (87.04 - 95.86) | 45.49 (43.53 - 47.52) | 29.25 (28.00 - 30.51) | 20.66 (19.82 - 21.52) |
> |                | LURE-Llama2-7B Dynamic | 63.25 (36.23 - 96.67) | 22.79 (14.56 - 32.91) | 13.52 (8.93 - 18.80) | 8.62 (5.82 - 11.88) |
> |                | LURE-Llama2-7B Fixed   | 42.71 (40.32 - 45.11) | 19.99 (19.07 - 20.96) | 12.42 (11.87 - 13.00) | 8.74 (8.38 - 9.11) |
> | FPB    7B-few | Uniform random      | 135.75 (130.00 - 141.72) | 65.14 (62.37 - 68.05) | 40.06 (38.41 - 41.76) | 27.99 (26.84 - 29.16) |
> |                | LURE-Llama2-7B Dynamic   | 65.73 (45.79 - 86.86) | 36.40 (25.31 - 49.18) | 18.33 (11.82 - 25.79) | 11.53 (7.37 - 16.43) |
> |                | LURE-Llama2-7B Fixed     | 77.54 (73.39 - 81.94) | 36.88 (35.22 - 38.61) | 22.97 (21.99 - 23.95) | 16.26 (15.56 - 16.96) |
> | HS     7B-few | Uniform random      | 156.59 (152.02 - 161.10) | 96.51 (93.78 - 99.32) | 77.32 (75.14 - 79.54) | 66.84 (64.93 - 68.76) |
> |                | LURE-Llama2-7B Dynamic | 121.45 (100.99 - 142.89) | 75.61 (60.04 - 91.73) | 52.12 (40.48 - 64.84) | 43.83 (35.35 - 52.63) |
> |                | LURE-Llama2-7B Fixed    | 144.44 (139.38 - 149.87) | 89.65 (87.31 - 91.98) | 72.62 (70.73 - 74.47) | 63.67 (62.07 - 65.31) |
> | Subj   7B-few | Uniform random      | 46.97 (44.97 - 49.00) | 21.79 (20.89 - 22.70) | 13.76 (13.19 - 14.36) | 10.23 (9.80 - 10.66) |
> |                | LURE-Llama2-7B Dynamic | 71.76 (50.93 - 95.54) | 35.25 (20.62 - 53.32) | 36.75 (21.11 - 55.19) | 30.94 (18.43 - 46.09) |
> |                | LURE-Llama2-7B Fixed     | 40.20 (38.39 - 42.02) | 19.69 (18.89 - 20.48) | 12.95 (12.42 - 13.50) | 9.12 (8.73 - 9.52) |
>
>
>
> | Acquisition step  |                   | 10    | 20    | 30    | 40    |
> | -----------------|------------------ | ----- | ----- | ----- | ----- |
> | SST-2 70B-few  | Uniform random      | 81.75 (77.77 - 85.85) | 40.04 (38.24 - 41.87) | 25.01 (23.98 - 26.06) | 17.92 (17.19 - 18.66) |
> |                | LURE-Llama2-7B Dynamic | 28.76 (22.13 - 35.70) | 17.21 (11.42 - 23.89) | 12.62 (8.83 - 16.74) | 8.57 (6.37 - 11.07) |
> |                | LURE-Llama2-7B Fixed   | 40.09 (38.28 - 41.98) | 19.11 (18.35 - 19.89) | 12.26 (11.80 - 12.73) | 8.44 (8.12 - 8.76) |
> | FPB    70B-few | Uniform random      | 173.90 (164.57 - 183.50) | 80.36 (76.55 - 84.22) | 50.14 (47.77 - 52.54) | 35.07 (33.40 - 36.83) |
> |                | LURE-Llama2-7B Dynamic   | 114.11 (84.88 - 144.11) | 34.89 (25.00 - 46.22) | 19.71 (13.20 - 27.05) | 11.10 (7.68 - 14.95) |
> |                | LURE-Llama2-7B Fixed     | 150.45 (140.12 - 161.06) | 70.59 (66.56 - 74.69) | 45.85 (43.59 - 48.21) | 33.25 (31.64 - 34.85) |
> | HS     70B-few | Uniform random      | 98.22 (94.33 - 102.28) | 48.73 (46.95 - 50.54) | 33.04 (31.84 - 34.29) | 25.27 (24.36 - 26.20) |
> |                | LURE-Llama2-7B Dynamic | 88.73 (64.35 - 116.85) | 41.15 (31.37 - 51.26) | 23.93 (18.57 - 29.54) | 16.18 (12.40 - 20.03) |
> |                | LURE-Llama2-7B  Fixed    | 74.57 (71.50 - 77.75) | 39.31 (37.78 - 40.87) | 26.87 (25.85 - 27.89) | 19.93 (19.18 - 20.69) |
> | Subj   70B-few | Uniform random    | 109.53 (105.47 - 113.54) | 66.96 (64.75 - 69.11) | 51.41 (49.77 - 53.10) | 42.97 (41.59 - 44.36) |
> |                | LURE-Llama2-7B Dynamic | 209.81 (161.39 - 260.62) | 134.48 (99.07 - 172.06) | 72.61 (54.29 - 92.06) | 64.43 (48.04 - 82.24) |
> |                | LURE-Llama2-7B Fixed     | 134.87 (129.79 - 140.22) | 77.31 (74.56 - 80.06) | 57.14 (55.20 - 59.07) | 47.24 (45.64 - 48.81) |
>
>
>
> > The authors conclude that the surrogate and target models do not have to be from the same family. However, the experiments only used two model families, so this result may not be very general. Additionally, the advantage of using a same-family surrogate model versus a different-family surrogate model was not quantified. This may be an area for future work.
>
> That’s a good point, thank you for giving us the opportunity to clarify! Our experiments actually span three model families, namely Llama-2, Gemma-3 and Phi-2 but some of these are only in the supplement (see Figures C.1. and C.2.). Evaluations demonstrate the robustness of active testing across model families.
> We agree that further investigation of the effects of model diversity between the surrogate and target would be interesting future work but are happy to offer some thoughts here: Interestingly, while the original active testing paper suggests that surrogate-target diversity is generally beneficial, we find that using smaller surrogates from the same model family can be highly effective. Although, there may still be scenarios where diversity between the target and surrogate model is more advantageous, it is possible this is less pronounced for LLMs, which are likely much more correlated in their predictions (e.g. due to shared architecture and parts of the training data) than the models considered by Kossen et al. (2021).
>
> > How were the datasets for the experiments selected?
>
> The datasets we chose are common LLM tasks, especially in the in-context learning literature (see In-context learning learns label relationships but is not conventional learning, Kossen et al, 2022 from which we took direct inspiration). We added MMLU as a more challenging, widely-used benchmark to demonstrate the robustness of active testing.
>
> > The authors ran experiments using datasets that are relatively low-difficulty for recent LLMs in 2025, and did not include datasets requiring open-ended generation. Open-ended generation datasets would have been a more realistic reflection of the problem setting, because these datasets are expensive to acquire human labels for.
>
> Evaluation on tasks requiring open-ended generation represents an interesting direction for future work, but we note that this faces fundamental technical challenges that extend beyond the scope of this submission. The acquisition functions for active testing that we study require the computation of the models predictive uncertainty. For open-ended generations with LLM, uncertainty quantification is still an area of active research (see for example Detecting hallucinations in large language models using semantic entropy, Farquhar et al, 2024), in particular when considering cost-effective uncertainty estimation.
>  As an alternative, we would like to point out that our classification tasks and especially MMLU remain challenging for the selection of models we studied, as shown in tables C7 and C8 of the supplementary material.
>
> > The graphs are too dense and hard to read.
>
> Thank you for pointing this out! We will revise the graphs accordingly to your suggestions and make sure they appear more clearly.
>
> Again, thank you for your review. We appreciate your questions and suggestions, which contribute to a more thorough analysis of our approach. Please let us know if there is anything else that we can clarify.

---

> > ### Comment · Reviewer_cFrJ · 2025-08-01
> >
> > Thank you for your detailed responses.
> >
> > ### Dynamic surrogate ablations
> >
> > Thank you for the additional experiments. The results indeed show that dynamic surrogates have no clear advantage. You might consider including the ablation experiments in your paper revision, as I believe it would strengthen your argument.
> >
> > ### Model families
> >
> > Thanks for your response. This adequately addresses my question.
> >
> > The idea that cross-family might be advantageous is interesting and may warrant future work. Would it be practical to run MMLU once with a _single_ surrogate model (e.g. Llama 3 8B), publish the predictive entropy for all instances as an open dataset, and then use this dataset for active testing for _any_ arbitrary large model? This would be a valuable resource for the community to have.
> >
> > ### Dataset selection, dataset difficulty, open-ended generation
> >
> > Thank you for your response. This adequately addresses my concerns. I acknowledge that applying this method to open-ended generations is difficult because of the lack of a reliable calibration method.
> >
> > I am curious if there are ways to work around this limitation. For instance, you could use sample multiple generations and then use the distribution of LLM-as-a-judge over those generations for risk estimation. But this could be quite expensive and impractical. Perhaps it could be an idea for future work.

---

> > > ### Author Response · Authors · 2025-08-06
> > >
> > > Thank you again for your thoughtful follow-up and continued enthusiasm for our work!
> > >
> > > > Thank you for the additional experiments. The results indeed show that dynamic surrogates have no clear advantage. You might consider including the ablation experiments in your paper revision, as I believe it would strengthen your argument.
> > >
> > > We are glad to hear that and will add these experiments in our paper revision.
> > >
> > > > The idea that cross-family might be advantageous is interesting and may warrant future work. Would it be practical to run MMLU once with a single surrogate model (e.g. Llama 3 8B), publish the predictive entropy for all instances as an open dataset, and then use this dataset for active testing for any arbitrary large model? This would be a valuable resource for the community to have.
> > >
> > > Thank you for this excellent suggestion! We agree that sharing surrogate predictions  on widely used benchmarks could be useful for practitioners and are considering adding this to future versions of the paper!
> > >
> > > > I acknowledge that applying this method to open-ended generations is difficult because of the lack of a reliable calibration method. I am curious if there are ways to work around this limitation. For instance, you could use sample multiple generations and then use the distribution of LLM-as-a-judge over those generations for risk estimation. But this could be quite expensive and impractical. Perhaps it could be an idea for future work.
> > >
> > > Thanks for the suggestion! We agree there is lots of interesting work to be done for uncertainty estimation in open-ended generation, including LLM-as-a-judge approaches.
> > >
> > >
> > > Thank you again for your hard work and, in particular, your engagement during the discussion period! We’re glad to see your continued support for our submission.

---

### Official Review · Reviewer_FX74 · 2025-06-29

**Clarity:** 3
**Significance:** 2
**Originality:** 1
**Rating:** 3
**Confidence:** 3

**Summary:**

This paper presents a scalable active testing method to make evaluating large language models more efficient. The approach overcomes prohibitive computational costs by using a smaller, fixed surrogate model, constructed cheaply via in-context learning, to intelligently guide the selection of data for labeling. By avoiding repeated training and minimizing the need for predictions from the large target model, this technique is shown to substantially reduce evaluation error compared to standard random sampling, enabling more accurate LLM assessment with less labeled data.

**Questions:**

Could the author expand more real world applications where labelling cost is not as important as inference compute time?

**Ethical Concerns:**

["NO or VERY MINOR ethics concerns only"]

**Final Justification:**

I think the word provides extensive results, because of this some reviewers find it valuable it could be accepted. However, my concern is that the idea and the paper itself is trivial in some sense. Most of acquisition are borrowed from previous work, and the one they proposed is basically assuming the surrogate behaves as well as the target model. Then if a small models performs well or have similar capabilities than the large model is obvious that we can use a small surrogate to do acquisition. This is not all the experiment they present, but many are around that idea.  It is possible that I am not weighting properly the contributions and that is why I put confidence 3.

**Limitations:**

yes

**Paper Formatting Concerns:**

.

**Quality:**

3

**Strengths And Weaknesses:**

**Strengths:**

- The method is clearly written, with a well-introduced overview of related work.

- The authors provide extensive evaluation.

**Weaknesses:**
- W1) Limited novelty: The central method primarily relies on approximating a target prediction with a cheaper prediction using ICL, an idea that by itself lacks novelty [1,2,3]. Additionally, the acquisition method presented in Section 3.3 resembles entropy-based acquisition methods commonly used in active learning. Philosophically, I do not know if we want to replace the induced test distribution by a cheap surrogate model over the specialized target model.

- W2) The proposed idea could be interesting if the ICL surrogate model effectively replaces a trained, task-specialized LLM. However, the evaluation predominantly tests only within the scope of in-context learning, and in the context of classification benchmarks. Thus, if the surrogate and target models exhibit similar capabilities—even with different model sizes (e.g., 7B and 70B)—particularly in certain domains, the findings become less compelling.
- W3) Consequently, if we assume small LLM to have similar capabilities to large LLMs, then the question is when this does not hold. I am a bit sceptic that the proposed approach "always" work. Since the paper is mostly empirical, I think we need to understand when does not work and when it work, which is not extensively explored.


### References

1. Diao, Shizhe, Pengcheng Wang, Yong Lin, Rui Pan, Xiang Liu, and Tong Zhang. “Active Prompting with Chain-of-Thought for Large Language Models.” *arXiv preprint* arXiv:2302.12246, 2023.

2. Zhang, Yiming, Shi Feng, and Chenhao Tan. “Active Example Selection for In-Context Learning.” In *Proceedings of the 2022 Conference on Empirical Methods in Natural Language Processing (EMNLP)*, pp. 9134–9148. Association for Computational Linguistics, 2022.

3. Margatina, Katerina, Timo Schick, Nikolaos Aletras, and Jane Dwivedi-Yu. “Active Learning Principles for In-Context Learning with Large Language Models.” In *Findings of the Association for Computational Linguistics: EMNLP 2023*, pp. 5011–5034. Association for Computational Linguistics, 2023.

---

> ### Author Rebuttal · Authors · 2025-07-30
>
> Thank you for your review. Unfortunately, we believe there may be a fundamental misunderstanding about our work’s scope and contributions. We hope to clarify these in our response below.
>
> Most importantly, we do not propose a method to improve model _predictions_ or to reduce their cost for standard inference applications. Our work only targets LLM _evaluation_. The core challenge is that evaluating frontier LLMs on large datasets is prohibitively expensive–both computationally and in terms of labeling costs. This is a critical bottleneck as models become larger and more numerous and as evaluation datasets expand.
>
> We do not claim novelty for in-context learning or entropy-based acquisition (W1), but rather demonstrate how to combine these techniques to make active testing practical for LLMs at scale. Our technical innovations aim to reduce the computational cost of active testing through (1) our use of fixed, in-context-learning–based, surrogate models, (2) using surrogates that are computationally cheaper than the target model, and, ultimately, (3) our proposal to replace even target model predictions with surrogate predictions for the largest computational cost savings.
>
>
> > W1) Limited novelty: The central method primarily relies on approximating a target prediction with a cheaper prediction using ICL, an idea that by itself lacks novelty [1,2,3]. Additionally, the acquisition method presented in Section 3.3 resembles entropy-based acquisition methods commonly used in active learning. Philosophically, I do not know if we want to replace the induced test distribution by a cheap surrogate model over the specialized target model.
>
> We believe there may have been a fundamental misunderstanding here that we hope is easy to clear up. Our use of the surrogate model does not “induce” a different test distribution, nor are we trying to improve on the predictions of the original model in any way: it is being used to construct a _proposal_ over which test points to acquire labels for, as is required by our R_LURE importance sampling estimator (Eq. 1). This estimator will be unbiased for any acquisition proposal distribution q that has full support (which ours do) and thus for any surrogate. For well-chosen q, the estimate with R_LURE can be significantly lower variance than a naive random sampling baseline. As the surrogate only affects the proposal, it does not introduce any difference in test distribution.  Importantly, the true labels and target model predictions always enter the estimator via the loss term, L(f(x), y), in the estimator, regardless of the choice for q.
>
> To reiterate, we do not suggest replacing target model predictions with surrogate predictions for general prediction scenarios. We only consider this for the purposes of computing the acquisition proposal q, as using a non-uniform proposal allows us to reduce the number of test labels we need to acquire, while using a small surrogate means this proposal is cheaper than using a large LLM. The estimator remains unbiased and estimates the test performance on the true distribution underlying the test data. It further retains the same convergence guarantees as a standard uniform Monte Carlo estimator.
>
> The works [1, 2, 3] that you refer to study how to actively construct the subset of samples to include in ICL, aiming to improve predictions. This is entirely orthogonal to our work as we do not actively construct the subset of samples used in the ICL of the surrogate. Instead, we use a fixed and randomly selected set of samples. Our acquisition function acquires samples for use in the active testing estimator and not for inclusion in ICL. We further note that Kossen et al 2021 have previously shown that active learning acquisition strategies are not directly optimal for the active testing estimator itself as the active learning and active testing problems are quite different in which points we wish to target.  As such, the fact that strategies related to entropy have been used in active learning is tangential to our actual contributions.
>
> We hope this clears up this misunderstanding and will clearly highlight differences and similarities to related work in a revised version of the draft.
>
> > W2) The proposed idea could be interesting if the ICL surrogate model effectively replaces a trained, task-specialized LLM. However, the evaluation predominantly tests only within the scope of in-context learning, and in the context of classification benchmarks. Thus, if the surrogate and target models exhibit similar capabilities–even with different model sizes– particularly in certain domains, the findings become less compelling.
>
> Again, this point suggests there may have been a fundamental misunderstanding of our method: our work has nothing to do with replacing the original “task-specialized LLM” with an ICL surrogate. As explained above, the surrogate is only used as a mechanism to more efficiently evaluate the original LLM by constructing an importance sampler.  Note also that it is a surrogate of the ground truth label distribution, not of the target’s distribution, with the aim to identify incorrect predictions of the target. Even a much smaller model (eg Gemma-3 4B or Llama-2 7B) can often identify errors made by a larger model (eg Llama-2 70B), because discrepancies in predictions can detect errors independently from task performance. Our experiments demonstrate this clearly, in Figures 2, 3, and Figures C1, C2 of the supplementary material. Whether the target model and final surrogate have similar capabilities when used in general prediction scenarios is irrelevant to our contribution, as we never suggest replacing the predictions of the former with the latter.
>
> > W3) Consequently, if we assume small LLM to have similar capabilities to large LLMs, then the question is when this does not hold. I am a bit sceptic that the proposed approach “always” work. Since the paper is mostly empirical, I think we need to understand when does not work and when it work, which is not extensively explored.
>
> We want to clarify that we never claim our approach always works independently from the surrogate’s choice, but this does not mean that we have to assume a small LLM has similar capabilities as a large LLM. We provide extensive experiments across a wide range of surrogates size, using Llama-2 7B and 70B, Gemma-3 4B and 1B as well as Phi-2 (note there are additional results in the appendix to investigate surrogate choice to those in the main paper), to analyze failure cases and boundary conditions. Our results demonstrate that active testing is generally beneficial when the surrogate is appropriately selected, even when the surrogate is a lot smaller than the target model, and fails gracefully in worst-case scenarios, with performance comparable to uniform sampling.  Even a small surrogate can provide meaningful signal in the acquisition function that allows us to improve over uniform random acquisition.
>
> > Could the author expand more real world applications where labelling cost is not as important as inference compute time?
>
> Computational costs dominate in several settings, such as model selection based on test datasets, or benchmark leaderboards when evaluating numerous models for comparison on large, already labeled datasets.
>
> Again, thank you for your review. We hope our response helped to clarify the motivation of our work and why its contributions are highly distinct from other uses of surrogates in an LLM context. Please let us know if there is anything else that we can clarify.

---

> > ### Comment · Reviewer_FX74 · 2025-08-07
> >
> > Thank you to the reviewers for their helpful feedback. After considering the concerns raised by other reviewers, I will maintain my original score.
> >
> > ---
> >
> > ### Influence of the Surrogate Model
> >
> > Thank you for clarifying that the underlying data distribution remains unchanged. However, the queried distribution does depend on the surrogate model’s capabilities, which directly affects data-collection efficiency. Experiment 5.5—where the target is removed entirely— illustrates this point. So the message that surrogate model do not have to have
> >
> > ### Novelty
> > After reading the authors’ response to Reviewer `x1Pz`, I remain unconvinced by the claimed novelty. The authors write:
> >
> > > “The idea of using an active approach for LLM evaluation is itself novel.”
> >
> > I respectfully think that applying an existing method to LLMs does not, in itself, constitute a substantial contribution. This is unless you are using a unique LLM capability, which could be  in-context learning  (ICL) as in this case. Demonstrating that an active-testing framework can operate with LLMs via is certainly valuable, but its novelty appears limited, since although in not the same context has been applied extensively.
> >
> > ### Other baselines
> > The authors assert:
> >
> > > “BALD methods are active-learning acquisition strategies, not active-testing acquisition strategies, as established in the original active-testing paper.”
> >
> > This distinction feels overly rigid for LLMs. Components of active-learning approaches—such as the predictive-entropy term from BALD—are often repurposed for model evaluation. As the authors note (L75–76), a strong acquisition function is crucial when surrogate and target models differ. When the models are similar, the acquisition function effectively reduces to measuring the target model’s entropy. This could be the case for 7 B and 70B LLMs if they are distilled from the they big model (which can be the case for many models). The position also appears inconsistent with Section 5.5 (L220–221), where surrogate-model entropy is used for acquisition.

---

> > > ### Author Response · Authors · 2025-08-07
> > > **Clarifications**
> > >
> > > Thank you for your continued engagement.  We would like to quickly make some clarifications on the new points you make.
> > >
> > > Firstly, the contributions of the paper extend beyond just directly applying the standard active testing framework to LLMs.  The approach laid out in the original active testing does not directly scale to working with LLMs and we have had to make various modifications (as laid out in the paper and various other discussions) to allow this.  We also feel there is significant novelty in the idea that LLMs can be evaluated in a very different way than virtually all work has done up to this point.  There is no previous mention or consideration of active testing within the LLM literature, so making this link is of significant utility to the community, even if the underlying idea is relatively simple.  Indeed, the **NeurIPS reviewer guidelines on novelty** say "originality does not necessarily require introducing an entirely new method" and can instead be from "offer[ing] a novel combination of existing techniques".  The ideas we are combining are currently very disparate and we feel our work clearly satisfies NeuRIPS' novelty criteria.
> > >
> > > Second, we would like to make clear that BALD is not even a possible strategy to use for the acquisition function here, regardless of whether it is deemed to be a sensible one or not: because there are no stochastic parameters in the model, we cannot target their expected information gain (i.e. the BALD score).  Similarly, most other popular active learning learning strategies like BADGE, BAIT, TypiClust, etc cannot be directly applied either. One strategy that could theoretically be applied is EPIG [1], but this would increase the cost by many orders of magnitude compared to our approach, thereby making it infeasbile at the LLM scale.  As such, while we would be happy to add an ablation to any concrete alternative acquisition strategy would like to see comparisons to and which can actually be run, we are not aware of any viable established alternatives that would give meaningfully distinct behaviour (e.g. it might be possible to construct margin-based acquisitions instead of entropy-based acquisitions, but this still falls under our general framework--equating to using a different loss in the equation at line 121--and is likely to give very similar behaviour, indeed it will be identical for binary classification).
> > >
> > > Thank you again for your continued engagement, but we do hope you reconsider your recommendation in light of the above.
> > >
> > > [1] Smith, F.B., Kirsch, A., Farquhar, S., Gal, Y., Foster, A. and Rainforth, T., 2023, April. Prediction-oriented bayesian active learning. In International conference on artificial intelligence and statistics (pp. 7331-7348). PMLR.

---

### Official Review · Reviewer_fX5b · 2025-07-01

**Clarity:** 4
**Significance:** 3
**Originality:** 3
**Rating:** 4
**Confidence:** 3

**Summary:**

This paper proposes an error estimation method, aiming to address the computational bottlenecks of active testing for LLMs. By sacling up current active testing methods, sampling-based active testing is chosen. It introduces lightweight surrogate models via in-context learning, eliminating costly iterative surrogate updates. The authors validate their method on Llama-2 and Gemma-3 models, experiments show that the proposed approach demonstrates lower estimation errorr compared to uniform random sampling across. They also introduce a single-run error estimator to assess the performance of active testing, which further improve effectiveness.

**Questions:**

1. In few-shot scenarios, why does the method fail more often than in zero-shot setups? Is it due to mismatches between surrogate and target model capabilities?
2. Can dynamic surrogate adaptation (e.g., updating surrogates with small label batches) mitigate errors in few-shot or complex tasks?

**Ethical Concerns:**

["NO or VERY MINOR ethics concerns only"]

**Final Justification:**

I have read authors' response. It addressing my concerns about the theoretical guarantees andI found explanation convincing.

While I appreciate the note on "graceful degradation" and agree it makes sense, the cases where performance lags behind random sampling still warrant deeper analysis. Exploring why the surrogate fails in these scenarios and including specific breakdowns of such failure cases would strengthen the work.

I plan to retain my positive rating.

**Limitations:**

See Weaknesses

**Quality:**

3

**Strengths And Weaknesses:**

## Strengths
1. Novel Idea: Instead of standard iterative training, the use of in-context learning for surrogate model construction and fixed-surrogate architecture drastically reduces computational overhead. It makes active testing feasible for LLM evaluations. It is really a good “scaling up” approach for active testing with LLM.
2. Novel Error Estimation: The single-run bootstrap estimator achieves high coverage of true risk errors, providing critical diagnostic tools for real-world deployment where multiple runs are infeasible.
3. Good Practical Impact: Error estimation is a critical problem for LLM, especially with limited computational and data budget. The framework’s applicability to dataset curation further extends its utility.
4.Easy to Follow: The paper is well-structured and easy to follow. Code availability and detailed implementation enhance reproducibility.

## Weaknesses
1. Theoretical Underpinnings: While empirically effective, the reliance on in-context learning for surrogate models may need more detailed theoretical analyses for generalization error bounds.
2. Instability in Few-Shot and Complex Tasks: Figure 2 for the 70B-few model on Subj and HS datasets, risk-estimation error is close to or worse than uniform random sampling. This may stem from surrogate models (trained via few-shot in-context learning) failing to capture the target model’s behavior when both are in few-shot modes. Figure 6  for the challenging MMLU dataset, active testing for 70B model indicating limits in surrogate model capability for complex tasks.

---

> ### Author Rebuttal · Authors · 2025-07-30
>
> Thank you for your hard work and helpful feedback! We are delighted that you appreciate the novelty and practical impact of our approach for cost-effective evaluation. We hope our response below alleviates your remaining concerns.
>
> > Theoretical underpinnings: While empirically effective, the reliance on in-context learning for surrogate models may need more detailed theoretical analyses for generalization error bounds.
>
> We respectfully disagree with the claim that the use of in-context learning for surrogate models significantly complicates obtaining theoretical guarantees: the surrogate is only being used as a proposal within a static importance sampling framework, so we inherit all the well established strong guarantees of unbiased i.i.d. Monte Carlo estimators (noting that our proposal is constructed in such a way that it is guaranteed to form a valid proposal so the estimator is unbiased and consistent). The guarantees will all be effectively the same as when using uniform sampling of new datapoints, except with a single sample variance. If our proposal is better than a uniform, this variance will be lower (and this is typically the case as shown in our experiments), if it is worse the variance will be higher.
>
>
> > Instability in Few-Shot and Complex Tasks: Figure 2 for the 70B-few model on Subj and HS datasets, risk-estimation error is close to or worse than uniform random sampling. This may stem from surrogate models (trained via few-shot in-context learning) failing to capture the target model’s behavior when both are in few-shot modes.
>
> While it is indeed an important point that the method can do similar or worse to uniform sampling if the surrogate performs poorly (and we have tried to ensure this is clearly shown and discussed in the paper), we emphasise that these are more graceful degradations rather than sudden instabilities.
>
> In particular, even in these worst-case scenarios, our method gracefully degrades to uniform sampling performance, maintaining a consistent –and close to 1– relative error. This highlights the robustness of our method, and is crucial for practical deployment as practitioners can safely apply our method without risk of obtaining estimations that are significantly worse than random sampling. We also note that such failures correspond to poorly capturing the true label distribution, rather than the target model’s behaviour itself.  We would be happy to add further clarifications, discussions, and/or ablations on this if you think it would improve the paper.
>
> > In few-shot scenarios, why does the method fail more often than in zero-shot setups? Is it due to mismatches between surrogate and target model capabilities?
>
> Excellent question! The key insight is that while the surrogate model uses few-shot learning in both scenarios, the target model receives different contexts. In zero-shot settings, the target model is not given any context and therefore performs worse, making more errors that can be easily identified by the surrogate model. In few-shot settings, the target model’s improved performance reduces the number of errors and the errors that do occur can be harder to predict, making it harder for the surrogate to identify them.
>
> > Can dynamic surrogate adaptation (e.g., updating surrogates with small label batches) mitigate errors in few-shot or complex tasks?
>
> Thank you for raising this important question! To test it, we explored surrogate adaptation and ran experiments with dynamic surrogate updating (detailed in our response to reviewer cFrJ). We didn’t observe clear consistent benefits in performance, but doing it drastically increased the computational costs: updating surrogates requires recomputing predictions over the entire sampling set with increasingly large contexts, which can be prohibitively costly in practice. We observed that experiments with surrogate updating took over 1500 times longer to run for 40 steps than when freezing surrogates.
>
> Given the huge expenses associated with this uncertain performance gain, we don’t believe surrogate adaptation is beneficial to active testing in most cases. However, you are right that there might be specific applications where adaptation could be valuable, particularly when labeling costs massively exceed computational costs. We view this as an interesting direction for future work and will add discussions on it along with our new results.
>
> Again, thank you for your review! We appreciate your questions and suggestions, which contribute to a more thorough analysis of our approach. Please let us know if there is anything else that we can clarify.

---

> > ### Comment · Reviewer_fX5b · 2025-08-05
> >
> > Thank you for addressing my concerns about the theoretical guarantees—I find your explanation convincing.
> >
> > While I appreciate the note on "graceful degradation" and agree it makes sense, the cases where performance lags behind random sampling still warrant deeper analysis. Exploring why the surrogate fails in these scenarios and including specific breakdowns of such failure cases would strengthen the work.
> >
> > I plan to retain my positive rating.

---

> > > ### Author Response · Authors · 2025-08-06
> > >
> > > Thank you for your positive feedback and careful assessment of our rebuttal!
> > >
> > > > While I appreciate the note on "graceful degradation" and agree it makes sense, the cases where performance lags behind random sampling still warrant deeper analysis. Exploring why the surrogate fails in these scenarios and including specific breakdowns of such failure cases would strengthen the work.
> > >
> > > We agree that failure cases analysis strengthens the work and have actually conducted such an analysis in Section 5.8. We analyse the SST-2 failure with 70B-few and identify a high rate of incorrect labels as the root cause, thereby demonstrating that some apparent “failures” are arguably more reflective of dataset quality issues rather than active testing limitations. Table C1 of the supplementary material provides further insight into a (graceful) failure case of active testing: we show that the correlation between our cross-entropy acquisition function and the negative-log likelihood of the target’s predictions (optimal acquisition function) is close to 0 when evaluating 70B-few on Subjectivity, which could explain degradation to uniform sampling performance.
> > >
> > > We will highlight these results more clearly in the next version of the draft. While we hope our response has already alleviated your remaining concern, please do let us know if you have any concrete suggestions for further analysis we could run, as we would be happy to include these in the next iteration of our paper. We appreciate your work as a reviewer and are glad to see your positive rating of our submission!

---

### Official Review · Reviewer_x1Pz · 2025-07-09

**Clarity:** 3
**Significance:** 2
**Originality:** 2
**Rating:** 3
**Confidence:** 4

**Summary:**

The paper addresses the problem of the increasing cost of evaluating frontier LLMs. The authors revisit active testing—which selectively sends the most informative test inputs for human-labelling—and show that three simple tweaks make it practical at today’s model scale: (1) build a surrogate model through in-context learning instead of gradient training; (2) replace the target model predictions with the (smaller) surrogate model predictions; and (3) when budgets are tight, for acquisition function computation, skip target model forward pass and instead rely only on the surrogate’s entropy. Combined with a bootstrap variance estimator that works “in one shot”, these changes cut the risk-estimation error by 32% (on avg.) compared with uniform random sampling across five text-classification benchmarks and two model families.

**Questions:**

1. The paper does not justify its choice of datasets; outlining the selection criteria and explaining how these datasets reflect the broader evaluation landscape would strengthen the empirical study.

2. How small can the surrogate model be before performance deteriorates, and how rapidly does evaluation accuracy decline as the surrogate’s quality decreases?

3. Can the proposed approach be extended to regression tasks with LLMs, and what modifications would that require?

4. How will this approach perform under distribution shifts?

**Ethical Concerns:**

["NO or VERY MINOR ethics concerns only"]

**Final Justification:**

Authors have addressed some of my concerns. However, my concerns regarding the novelty of the work still remain. So, I still tend towards rejection (I have raised my score to 3).

**Limitations:**

Although the authors have discussed some limitations, the paper would benefit from a deeper analysis of the conditions under which the proposed estimators are reliable—and where they may fail—as well as a clearer discussion of how far the empirical findings can be generalized beyond the specific datasets and models tested.

**Paper Formatting Concerns:**

No major concerns regarding the paper formatting.

**Quality:**

2

**Strengths And Weaknesses:**

**Strengths**

1. The paper tackles an important problem and is both clearly written and well structured.

2. The proposed method yields substantial computational savings for LLM evaluation and remains flexible across settings and models.

**Weaknesses**

1. The acquisition rule—uncertainty sampling via Shannon entropy—and the use of surrogate models are already commonplace in active-learning research (e.g., Settles 2010; Dongyuan et al., 2024), limiting methodological novelty.

2. Freezing the surrogate model will prevent the active sampler from incorporating information gained during data collection, which can degrade efficiency.

3. Uniform random sampling is a weak baseline; comparisons against stronger alternatives such as margin sampling, BALD etc. are needed to gauge true gains.

4. There are no theoretical guarantees presented for the recommended estimators (using surrogate model), leaving their performance behaviors under different settings unclear.

**References**

1. Active Learning Literature Survey by Burr Settles, 2010

2. A Survey on Deep Active Learning: Recent Advances and New Frontiers by Dongyuan et. al. 2024

---

> ### Author Rebuttal · Authors · 2025-07-30
>
> Thank you for your feedback! We appreciate that you recognize the substantial computational savings and flexibility of our method. We hope our response below alleviates your concerns.
>
> > The acquisition rule–uncertainty sampling via Shannon entropy–and the use of surrogate models are already commonplace in active-learning research, limiting methodological novelty.
>
> There is a significant difference between active learning and active testing and our core novelty is being the first paper that shows how active testing can be successfully used in LLM evaluation.  We do not claim that the use of Shannon entropy is a novelty, nor the use of surrogate which was already introduced in the original active testing paper. Our technical contribution is the overall adaptation to active testing to make it practical for LLMs at scale. Technical innovations that have not previously been done in the active testing context include: (1) using in-context learning to create surrogates that are fixed during sampling, without expensive retraining; (2) using a significantly smaller surrogate compared to the target, thereby reducing inference costs for surrogate predictions; (3) replacing the target model predictions with surrogate predictions to reduce computational costs even further when budgets are tight.
>
> > Uniform random sampling is a weak baseline; comparisons against stronger alternatives such as margin sampling, BALD etc. are needed to gauge true gains.
>
> We disagree with the claim that uniform sampling is not an appropriate baseline, as it is the standard approach used in practice for evaluation.  The idea of using an active approach for LLM evaluation is itself novel, while approaches like margin sampling and BALD are active _learning_ acquisition strategies, not active _testing_ acquisition strategies. As established in the original active testing paper (Active testing: sample-efficient model evaluation, Kossen et al, 2021), most active learning acquisition functions are inappropriate for testing scenarios because active testing aims for accurate risk estimation rather than model improvement, which is conceptually a different problem. Our contribution lies in making active testing practical for LLMs despite prohibitive training and inference costs, demonstrating that existing acquisition functions can be useful for computational and labeling efficiency.
>
> > There are no theoretical guarantees presented for the recommended estimators (using surrogate model), leaving their performance behaviors under different settings unclear.
>
> We strongly disagree that our method lacks theoretical guarantees. Our approach is grounded in established importance sampling theory: we employ a static importance sampler that is provably unbiased (On statistical bias in importance sampling: How and when to fix it, Farquhar et al, 2021) and samples uniformly at random (i.i.d) since the surrogate distribution remains fixed. It therefore maintains the same theoretical guarantees as uniform sampling. In particular, following guidelines from Active testing: sample-efficient model evaluation, Kossen et al, 2021, our proposal is known to be valid, and thus its convergence and unbiasedness are guaranteed. Additionally, this estimator can drastically reduce the variance of estimates compared to uniform sampling when the acquisition proposal is properly chosen.
>
> We appreciate though that these theoretical properties could have been made clearer in the submission and will update it accordingly.
>
> > Although the authors have discussed some limitations, the paper would benefit from a deeper analysis of the conditions under which the proposed estimators are reliable—and where they may fail—as well as a clearer discussion of how far the empirical findings can be generalized beyond the specific datasets and models tested.
>
> The variety of experiments we conducted does in fact allow for  extensive analysis of when our estimators are reliable versus when they fail. Our estimators prove reliable when the surrogate can effectively identify target model errors. When this correlation breaks down, performance gracefully degrades to uniform sampling. This can happen when the surrogates are too small for the target’s complexity (see examples with Phi-2 and Gemma-3 1B in Figures C1 and C2 of the supplementary material), for challenging tasks where surrogates struggle to capture incorrect answers (see MMLU dataset in Section 5.7), or when labels are noisy (as discussed in Section 5.8). Even in failure modes, our method degrades to uniform sampling’s performance, providing estimates that a practitioner can still rely on.
>
> Our experiments span three model families, sizes from 1B to 70B parameters and diverse tasks. The consistent 32% average improvement across these settings demonstrates broader applicability. The key insight is that effectiveness depends on the acquisition proposal and therefore the surrogate’s ability to identify the target’s error rather than absolute capabilities, making the approach transferable to new models and tasks.
>
> > Freezing the surrogate model will prevent the active sampler from incorporating information gained during data collection, which can degrade efficiency.
>
> Our decision to freeze surrogate models is empirically justified by our experimental results, leading to significant gains in error estimation, and computationally motivated, saving additional costs. Updating surrogates requires recomputing predictions over the entire sampling set, with increasingly large contexts at each step, which defeats our motivation for computational efficiency.
>
> That said, we have now added a new ablation with dynamic updating of the surrogate to assess how big the benefits of doing this might be, see our response to reviewer cFrJ for the new results. These new results show that dynamically updating the surrogate does not provide clear consistent benefits, while prohibitively increasing computational costs: in our new experiments using dynamic surrogates for 40 steps only was over 1500 times more expensive than the frozen ones we recommend.
>
> > The paper does not justify its choice of datasets; outlining the selection criteria and explaining how these datasets reflect the broader evaluation landscape would strengthen the empirical study / Can the proposed approach be extended to regression tasks with LLMs, and what modifications would that require?
>
> The SST-2, FPB, Hatespeech and Subjectivity datasets are very common LLM tasks, especially in the in-context learning literature (see for example In-context learning learns label relationships but is not conventional learning, Kossen et al, 2022 from which we took direct inspiration for our experiment selection). We added the more challenging MMLU dataset to confirm our results on a commonly used and more difficult classification task. While our method extends naturally to any task type, as the acquisition function can be anything, as explained in Section 2, LLM evaluation predominantly focuses on classification due to the discrete token space. With classical neural networks, Kossen et al. 2021 (Active Testing: Sample-Efficient Model Evaluation) have already explored application of active testing to regression tasks, where the acquisition function now approximates MSE.
>
> > How small can the surrogate model be before performance deteriorates, and how rapidly does evaluation accuracy decline as the surrogate's quality decreases?
>
> Good question! In general, active testing provides bigger benefits the better the surrogate is. Our analysis of small surrogates (Phi-2 in Figure C1 and Gemma-3 1B in Figure C2 of the supplementary material) identifies the practical size limits for surrogate models for our selection of tasks. For some tasks, we observe performance deteriorating when these small models evaluate much larger targets like Llama-2 70B-few. Importantly, active testing does not fail catastrophically for decreasing surrogate quality. Instead, it gracefully degrades towards the label-efficiency of standard uniform sampling. Note however, that small models do outperform uniform sampling in many settings, including when evaluating Llama-2 7B.
>
> While these results provide empirical evidence demonstrating the robustness of active testing under deteriorating surrogate quality, we agree it would be interesting work to further quantify this. We note, however, that any useful measurement of ‘surrogate quality’ would need to take into account the target model and evaluation task – and is thus unlikely something a practitioner could compute upfront. This is where we see our bootstrap estimates providing valuable insight for practitioners applying active testing!
>
>
> > How will this approach perform under distribution shifts?
>
> Given that the models we used haven’t directly been trained on these tasks, we are not sure one can really think of things in terms of a “distribution shift” so much as there being tasks the LLM finds relatively harder to predict than others. We test on tasks of varying difficulty and find our approach to be helpful on both easier and harder tasks, indicating that the approach remains robust to this kind of problem.
>
>
> Again, thank you very much for your review. We hope that our response has addressed your concerns. Please let us know if there’s anything else we can clarify.

---

> > ### Comment · Reviewer_x1Pz · 2025-08-06
> >
> > Thanks for your detailed response and answering my questions, specifically
> >
> > 1) Conducting additional experiments with dynamic surrogate,
> >
> > 2) Clarifying concerns about the baseline, and
> >
> > 3) Clarifying the main contributions of the paper.
> >
> > However, I still tend towards rejection because of the concerns regarding the novelty of the work (I will raise my score to 3).

---

> > > ### Author Response · Authors · 2025-08-06
> > > **Author Reply to Reviewer x1Pz**
> > >
> > > Thank you for your positive feedback regarding our additional experiments with dynamic surrogates. We are also glad to hear you appreciate our clarifications regarding the baseline validity and the main contributions this paper makes to address the important problem of LLM evaluation.
> > >
> > > > However, I still tend towards rejection because of the concerns regarding the novelty of the work.
> > >
> > > We would love to discuss these concerns in more detail in the remainder of the discussion period – and clear them up if possible, as we believe they are based on a misunderstanding of our contributions.
> > >
> > > As we have already mentioned in our rebuttal above, we do not claim novelty for using entropy-based acquisition or surrogate models – not least because both are already an integral part of the original active testing framework. Instead, our technical contributions come from adapting the active testing framework to make it practical for use with LLMs at scale. This necessitated a variety of significant changes, including (1) using in-context learning to create surrogates that are fixed during sampling, without expensive retraining; (2) using a significantly smaller surrogate compared to the target, thereby reducing inference costs for surrogate predictions; (3) entirely replacing the target model predictions with surrogate predictions to reduce computational costs even further when budgets are tight. All of these present a significant departure from previous guidance for active testing with general purpose models, with (1-2) directly opposing recommendations in the original active testing publication by Kossen et al. (2021).
> > >
> > > We believe that the research community and practitioners will benefit from the publication of our findings, and would very much appreciate your support of our submission. Thank you again for your engagement during the discussion period so far!

---

### Decision · Program_Chairs · 2025-09-17

**Decision:**

Accept (poster)

**Comment:**

This paper adapts active testing and active surrogate estimators to evaluate LLMs, addressing the challenge of costly label acquisition for large test sets.
While the methods are based on the combinations of the known established techniques, their integration and application to LLMs represent a novel contribution.
The authors also provided some discussion on the failure cases analysis of their approach, which further strengthen the paper.